# Endogenous cannabinoids in the piriform cortex tune olfactory perception

Geoffrey Terral [1,2], Evan Harrell [1,6], Gabriel Lepousez [3,6], Yohan Wards [2], Dinghuang Huang [2], Tiphaine Dolique[4], Giulio Casali [1], Antoine Nissant [3], Pierre-Marie Lledo[3], Guillaume Ferreira [5], Giovanni Marsicano [2,7] & Lisa Roux [1,7] ✉

Sensory perception depends on interactions between external inputs transduced by peripheral sensory organs and internal network dynamics generated by central neuronal circuits. In the sensory cortex, desynchronized network states associate with high signal-to-noise ratio stimulus-evoked responses and heightened perception. Cannabinoid-type-1-receptors (CB1Rs) - which influence network coordination in the hippocampus - are present in anterior piriform cortex (aPC), a sensory paleocortex supporting olfactory perception. Yet, how CB1Rs shape aPC network activity and affect odor perception is unknown. Using pharmacological manipulations coupled with multi-electrode recordings or fiber photometry in the aPC of freely moving male mice, we show that systemic CB1R blockade as well as local drug infusion increases the amplitude of gamma oscillations in aPC, while simultaneously reducing the occurrence of synchronized population events involving aPC excitatory neurons. In animals exposed to odor sources, blockade of CB1Rs reduces correlation among aPC excitatory units and lowers behavioral olfactory detection thresholds. These results suggest that endogenous endocannabinoid signaling promotes synchronized population events and dampen gamma oscillations in the aPC which results in a reduced sensitivity to external sensory inputs.

Sensory perception depends not only on the transduction of physical phenomena like light, pressure, and volatile chemical species in the outside world but also on internally generated network dynamics. In the sensory cortex, desynchronized network states – or activated states - are typically associated with enhanced reliability of sensory-evoked responses and heightened perception as compared to states with strong neuronal synchronization[1–7]. In the neocortex, the alternations between low and high synchronization states have been shown to rely on neuromodulators such as acetylcholine[8] and on thalamo-cortical loops, which play a key role in neuronal coordination[9]. Despite the importance of network states and neuronal synchronization for

sensory processing[10,11], it remains unclear whether and how such phenomena are manifested in the olfactory piriform cortex, a sensory paleocortex that does not receive direct thalamic inputs like the sensory neocortex[12], but does show odor-evoked responses[13–16] and plays a key role in olfactory perception[17–19].

The anterior piriform cortex (aPC) is a large cortical area that receives strong unstructured inputs from the main olfactory bulb[12,20,21]. As such, it is thought to primarily encode the sensory features of odorants but also "odor objects"[19]. Because the aPC contains a dense network of association fibers connecting excitatory principal cells across relatively long distances[12,22], its circuits are well suited to study

[1]Univ. Bordeaux, CNRS, Interdisciplinary Institute for Neuroscience, IINS, UMR 5297, F-33000 Bordeaux, France. [2]Univ. Bordeaux, INSERM, Neurocentre Magendie, U1215, F-33000 Bordeaux, France. [3]Perception and Memory Unit, CNRS, Joint Research Unit 3571, Université Paris Cité, Institut Pasteur, 75015 Paris, France. [4]Inovarion, 75005 Paris, France. [5]Univ. Bordeaux, INRAE, Bordeaux INP, NutriNeurO, UMR 1286, F-33000 Bordeaux, France. [6]These authors contributed equally: Evan Harrell, Gabriel Lepousez. [7]These authors jointly supervised this work: Giovanni Marsicano, Lisa Roux. ✉e-mail: lisa.roux@u-bordeaux.fr

the interactions between external inputs and internal circuit dynamics. Seminal work from Hasselmo and colleagues showed, for instance, that acetylcholine can modulate the balance between external and internal influences by suppressing the intra-cortical associative connections through the activation of presynaptic muscarinic receptors[23]. Besides this well-characterized regulation of piriform circuits by acetylcholine, the function of other neuromodulators in this brain region has been studied[24]. Yet, how the endocannabinoid system (ECS) modulates aPC circuit function in vivo remains elusive[25].

By activating the main effectors of the ECS, the type-1 cannabinoid receptors (CB1Rs), endogenous (i.e., endocannabinoid) and exogenous (i.e., plant-derived or synthetic agonists) ligands impact diverse cognitive functions[26-28]. For instance, it was established more than 50 years ago that cannabis consumption is associated with altered sensory perception[29]. Yet, besides this effect induced by *exogenous* CB1R ligands, little is known about the physiological actions of endocannabinoids in sensory processing, including in olfaction[25,30].

In the aPC, CB1Rs are mainly expressed in the axon terminals of GABAergic neurons[31] that project to layers 2 and 3, where they control endocannabinoid-mediated plasticity[32]. CB1Rs are also strongly expressed in aPC-to-OB corticofugal axon terminals[33]. The physiological role of CB1Rs in piriform synaptic transmission and their impact on specific olfactory-guided behaviors has been explored previously[31,32,34,35]. However, little is known about the functions of endocannabinoid signaling at the network level. In fact, studies testing the impact of endogenous activation of CB1Rs in the physiology of sensory systems in vivo appear to be limited (but see ref. 36). As such, the specific impact of endocannabinoid signaling on aPC network coordination has not been investigated yet, despite the fact that it is altered by exogenous CB1R activation in other brain regions such as the hippocampus[37-39]. Indeed, hippocampal sharp-wave ripples and unit entrainment by the theta rhythm is disrupted with CB1R manipulations[37,40]. Considering the importance of synchronization states in sensory processing[10,11], we hypothesized that CB1Rs could tune olfactory perception via their impact on aPC network dynamics.

To test the impact of endocannabinoids on aPC network activity and on odor detection abilities we used a combination of pharmacology, olfactometry, silicon probe, and fiber photometry recordings in the aPC of freely behaving mice. Our study highlights a role for endocannabinoids in physiological conditions whereby they facilitate short-timescale synchronization among neuronal circuits at the expense of gamma oscillations. As mice with CB1R blockade in the aPC show heightened abilities to detect low-concentration odorants, we propose that the endocannabinoids in the aPC, by tuning the internal network dynamics, dampen olfactory perception.

## Results

### Systemic CB1R blockade disrupts spike timing coordination in the aPC

To study whether the endocannabinoid system regulates aPC circuits, we administered the CB1R antagonist Rimonabant (Rim, 1 mg/kg, intraperitoneal, i.p.) to freely moving mice implanted with silicon probes in the aPC (Fig. 1a, b). Neuronal activity (units and local field potentials) was recorded from 17 sessions of Vehicle- and 11 sessions of Rim-treated animals as they were awake in their home cage (9 and 8 mice for Vehicle and Rim conditions, respectively). Based on the waveform features of the recorded units[41] and putative monosynaptic connections assessed using cross-correlated spike trains of unit pairs[42,43], we evaluated that 717 were putative excitatory (E_Cells) and 102 were putative inhibitory units (I_Cells) (Supplementary Fig. 1). CB1R manipulations in other brain regions have been associated with dysregulation of neuronal synchrony. For instance, knocking out CB1Rs in the secondary visual cortex alters correlated activity among neurons[36]. Activation of CB1R also reduces the occurrence of hippocampal sharp-wave ripples (SWRs) and disturbs

the spiking coordination of neurons in the hippocampus and prefrontal cortex[37,39]. Similar to the hippocampus, aPC generates synchronous network events (population events, PopEvents) reminiscent of SWRs[44-46]. We detected these events based on the increased firing rate of E_Cells during short (~150 ms) time windows (Methods; Fig. 1c, d). In order to test whether endocannabinoid signaling impacts synchronous network events in the aPC, we first investigated the effect of systemic CB1R blockade on these PopEvents and found that Rim injections decreased their occurrence rate (Fig. 1e, f). We also examined the impact of the CB1R antagonist on the spiking coordination of pairs of neurons. By computing the cross-correlograms of all pairs of E_cells, we observed that CB1R blockade impaired the coordination of the units mainly at short-time scales (cross-correlation integral [−50 to +50 ms], $P = 8.27e-22$, Fig. 1g; [−200 to +200 ms], $P = 2.70e-14$; [−1000 to −800 ms|+800 to 1000 ms], $P = 2.27e-07$; two-sided Mann–Whitney tests). Although PopEvents represented on average only 5.9% (±0.2 SEM) of the total time and 17.9% (±2.8 SEM) of the total number of recorded spikes in the considered periods ($n = 25$ sessions), the decreased correlation observed in Rim was absent when excluding PopEvents from the cross-correlogram analysis (Fig. 1h). This result indicates that the decrease in cross-correlation induced by CB1R blockade occurred primarily during PopEvents.

Measures of spike coordination at short-time scales can be influenced by firing rates, as lower rates lead to a decreased probability of observing two spikes in close proximity for a given neuronal pair[47]. To account for this possibility, we analyzed the impact of CB1R blockade on the activity patterns of individual units (Supplementary Fig. 2). While the effect of Rim on I_Cells was similar to Vehicle, we found that CB1R blockade increased the firing rates of E_Cells, yet with a small effect size (Cohen's $D$: 0.105). This observation indicates that the decreased coordination among E_Cells cannot be explained by reduced activity levels but instead results from changes in the temporal organization of the spikes. Altogether, these experiments revealed that endogenous activation of CB1Rs minimally affects the baseline activity of individual aPC units but promotes co-activation of E_Cells, specifically during PopEvents, thereby contributing to neuronal coordination in the aPC.

### Systemic CB1R blockade promotes gamma oscillations in the aPC

Neuronal oscillations play fundamental roles in olfactory circuits, with oscillations in the beta and gamma range being strongly linked to sensory perception and attentive states[48-50]. We examined the effect of CB1R blockade on aPC local field potential (LFP) oscillations in the theta (4–12 Hz), beta (12–30 Hz), and gamma (30–80 Hz) frequency bands. Systemic injections of Rim increased the power of the signal in all frequency bands but mainly in the frequency range corresponding to low gamma (Fig. 2a–c). The main olfactory bulb is known to produce powerful gamma oscillations[48-51], raising the possibility that gamma oscillations recorded in aPC are, in fact, derived from the bulb via volume conduction. To test for this possibility, we performed current source density analysis (CSD) on the recordings where our recording sites spanned layers I and II of the aPC (Supplementary Fig. 3a). We found the presence of a clear dipole between layer II and layer I with sources and sinks alternating within each gamma cycle, ruling out the possibility that aPC gamma is purely volume-conducted. To take this analysis further, we examined the spike-gamma phase-coupling of the recorded units. CB1R blockade did not change the phase preference of E_Cells (they were preferentially active during the ascending phase of the cycles—Fig. 2d, e), but it accentuated the entrainment of the units specifically by gamma oscillations, as reflected by an increase in the resultant vector length (a measure of the units' modulation strength) as compared to units in the Vehicle group (Fig. 2d, e).

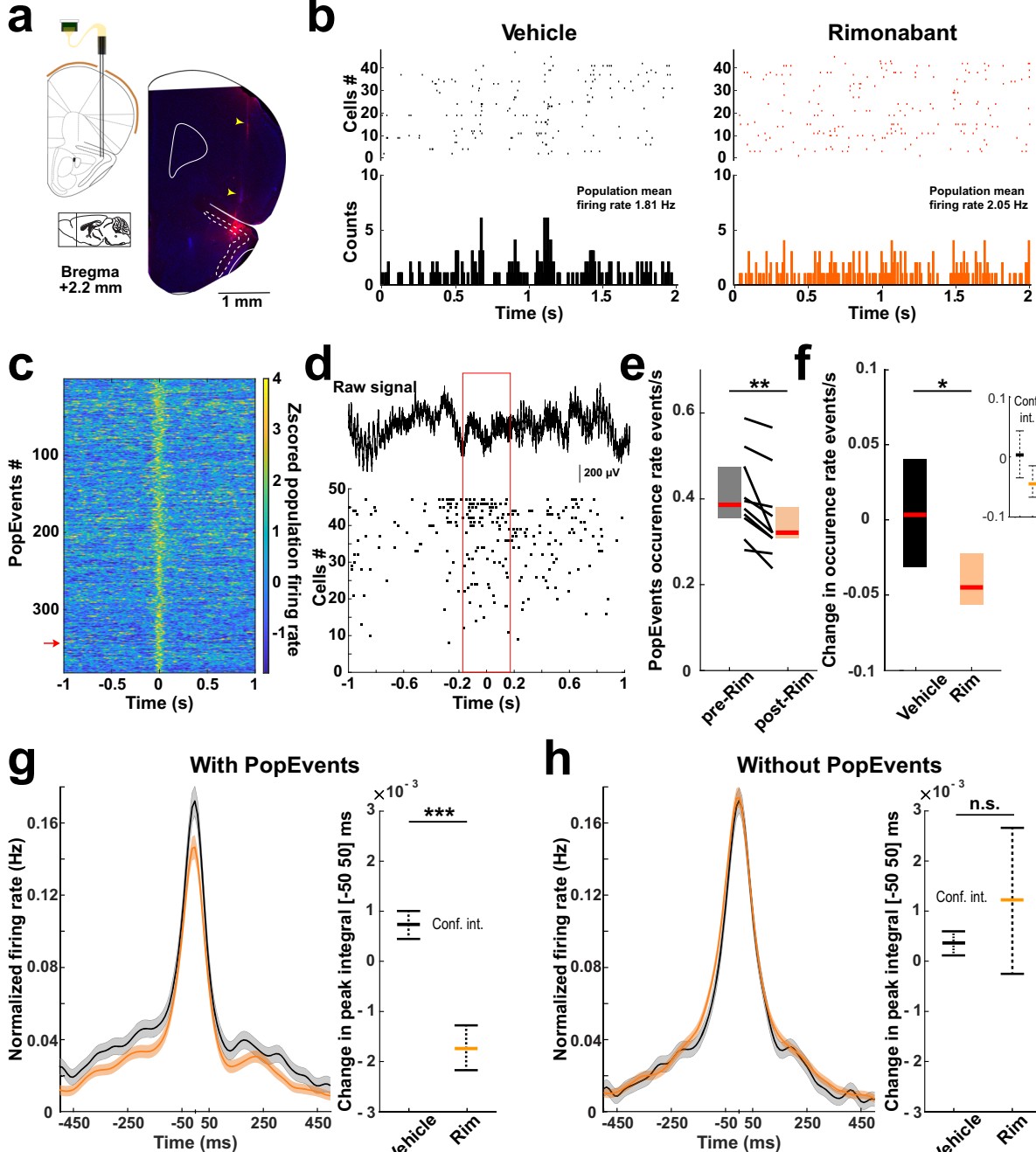

**Fig. 1 | Effect of CB1 receptor blockade on unit coordination in the anterior piriform cortex. a** Left: schematic of silicon probe implant in the anterior piriform cortex. Right: coronal section with electrolytic lesion and probe track (yellow arrows) (red: immunostaining against mouse antibodies; blue: DAPI). Probe placement was similarly verified in all recorded mice. Adaptation from Paxinos and Watson atlas. **b** Top: example of raster plots from putative excitatory units (E_Cells) following systemic injection of Vehicle (left) or CB1 receptor antagonist (Rimonabant–Rim, right). Bottom: spike counts (10 ms bins). **c** Z-scored convolved spike trains for E_Cells aligned on individual population events (PopEvents) of an example session. **d** Top: example of layer 2 raw signal during a PopEvent (red arrow in **c**). Bottom: corresponding E_Cells raster plot (red box: PopEvent interval). **e** PopEvents occurrence rate, pre- and post-injections of Rim. pre-Rim: 0.39 [0.36 0.48] events/s; post-Rim: 0.32 [0.31 0.38] events/s. Two-sided Wilcoxon's test: **$P = 0.002$; $n = 10$ sessions in 7 mice. **f** Median change in PopEvent occurrence rate

following Vehicle and Rim injections. Vehicle: 0.003 [−0.031 0.040]; Rim: −0.045 [−0.056 −0.022] events/s. Two-sided Mann–Whitney test: *$P = 0.02$; $n = 15$ and 10 sessions from 8 and 7 mice for Vehicle and Rim, respectively. **g** Left: Mean cross-correlograms of E_Cell pairs, pre- and post-injection of Rim, when including PopEvent firing. Means, bold lines; SEM, shaded areas. Right: median change of peak integral of the cross-correlograms from −50 to 50 ms. Vehicle 0.73 [−6.60 9.44] ($\times 10^{-3}$); Rim −1.74 [−13.13 10.06] ($\times 10^{-3}$). Two-sided Mann–Whitney test: ***$P = 8.27e{-22}$; $n = 6764$ and 5099 pairs from 17 sessions (9 mice) and 11 sessions (8 mice) for Vehicle and Rim, respectively. **h** Same as **g** but excluding PopEvent activity. Vehicle 0.37 [−6.21 7.50] ($\times 10^{-3}$); Rim 1.23 [−27.27 29.98] ($\times 10^{-3}$). Two-sided Mann–Whitney test: $P = 0.78$; $n = 6015$ and 4787 pairs from 15 sessions (8 mice) and 10 sessions (7 mice) for Vehicle and Rim, respectively. Boxplots and values in legend represent median and 25–75th percentiles ([lower bound upper bound]). n.s. non-significant. Source data are provided as a Source Data file.

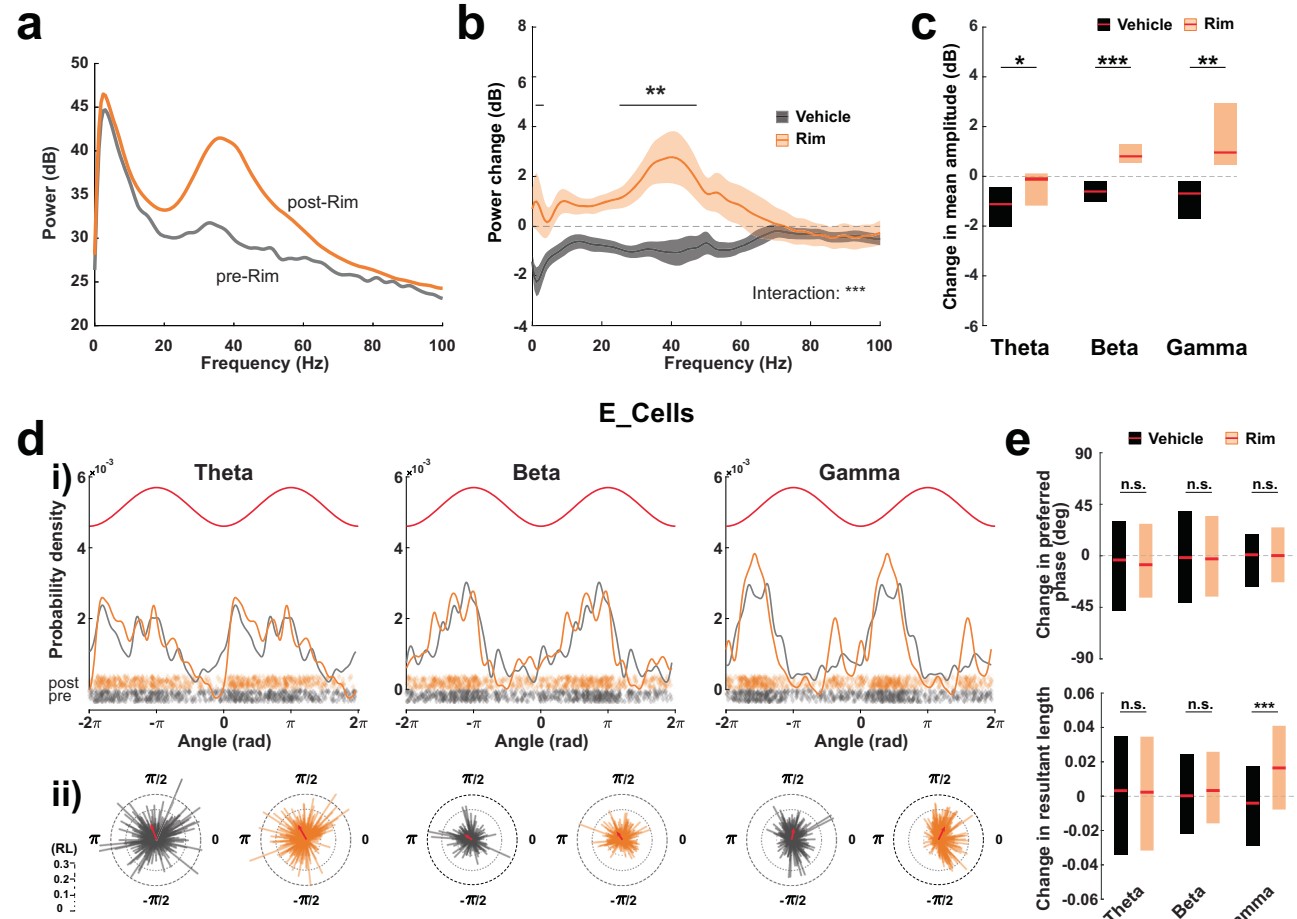

**Fig. 2 | Effect of CB1 receptor blockade on anterior piriform cortex oscillations.**
**a** Example of power spectra before (gray) and after (orange) systemic injection of Rimonabant (Rim). **b** Mean change in power spectrum induced by Vehicle (black) or Rim (orange). Means, bold lines; SEM, shaded areas. Two-way ANOVA interaction: ***$P < 0.001$. Horizontal line: post-hoc $t$ test with Bonferroni correction: *$P < 0.05$. $n = 17$ and 11 sessions from 9 and 8 mice for Vehicle and Rim, respectively. **c** Changes in mean amplitude of Theta-, Beta-, and Gamma-filtered signals. Theta: Vehicle −1.11 [−2.02 −0.42]; Rim −0.11 [−1.17 0.12] dB (*$P = 0.043$). Beta: Vehicle −0.60 [−1.03 −0.22]; Rim 0.81 [0.53 1.29] dB (***$P = 5.22e−05$). Gamma: Vehicle −0.68 [−1.70 −0.18]; Rim 0.96 [0.45 2.92] dB (**$P = 0.002$). $n = 17$ and 11 sessions from 9 and 8 mice for Vehicle and Rim, respectively. **d** (**i**) Probability densities of preferred phase for E_Cells, pre- (gray) or post- (orange) Rim injections. Top: theoretical cycles (red). Bottom (diamonds): preferred phase of individual units. (**ii**) Polar distributions of the preferred phases and resultant lengths (RL) for individual

units. Red: mean preferred phases and RL for the population. $n = 305$ units from 11 sessions (8 mice). **e** Top: Median change in preferred phase. Theta: Vehicle −3.66 [−47.93 29.83]; Rim −7.81 [−36.50 27.84] degrees ($P = 0.705$). Beta: Vehicle −1.63 [−40.59 38.74]; Rim −2.70 [−35.58 34.65] deg ($P = 0.990$). Gamma: Vehicle 0.87 [−26.52 18.81]; Rim 0.14 [−23.12 24.71] deg ($P = 0.241$). Bottom: Median change in RL. Theta: Vehicle 0.0032 [−0.034 0.035]; Rim 0.0023 [−0.032 0.035] ($P = 0.612$). Beta: Vehicle 0.00020 [−0.022 0.024]; Rim 0.0033 [−0.016 0.026] ($P = 0.166$). Gamma: Vehicle −0.0041 [−0.029 0.017]; Rim 0.016 [−0.0078 0.041] (***$P = 5.64e−16$). $n = 456$ and 305 E_Cells from 17 and 11 sessions in 9 and 8 mice for Vehicle and Rim, respectively. Two-sided Mann–Whitney tests were used to assess significance unless stated otherwise. Boxplots and values in the legend represent median and 25–75th percentiles ([lower bound upper bound]). n.s. non-significant. Source data are provided as a Source Data file.

The phase-coupling of I_Cells was also increased selectively for gamma oscillations following Rim injections, with a modest shift in their preferred phase toward earlier cycle phases when compared to Vehicle injections (Supplementary Fig. 3b, c). Besides this increased gamma coupling observed at the population level, the proportion of modulated E and I_Cells in aPC remained similar to pre-injection periods in Vehicle and Rim-injected mice (Rayleigh test, $P > 0.05$; 239 out of 456 modulated E_Cells in Vehicle, Chi-square test $P = 0.64$ when compared to pre-injection period; 174 out of 305 modulated E_Cells in Rim, Chi-square test $P = 0.87$; 43 out of 79 modulated I_Cells in Vehicle, Chi-square test $P = 0.75$; 45 out of 77 modulated I_Cells in Rim, Chi-square test $P = 1$). Altogether, these data suggest that endocannabinoid signaling down-regulates gamma oscillations and the strength of spike-gamma phase coupling of aPC units without changing the proportion of units entrained by gamma oscillations.

## Negative correlation between population events and gamma power changes under systemic CB1R blockade

Since CB1R blockade impacts both gamma oscillations and short-timescale neuronal co-activation in the aPC we wondered whether these two phenomena are functionally associated. To assess their potential relationship, for each session, we compared the change in gamma amplitude with the change in unit cross-correlation induced by Rim injections. We found a significant negative correlation between these two features: the higher the increase in gamma power, the stronger the reduction in neuronal co-activation at short-time latencies (cross-correlation peak integral, [−50 to +50 ms]; Supplementary Fig. 4a, b). This negative correlation was no longer present when excluding PopEvents from the cross-correlation analysis (Supplementary Fig. 4a, b), was specific for gamma oscillations, and was primarily observed for short latencies in cross-correlograms (Supplementary Fig. 4b). Considering that CB1R blockade affects neuronal co-firing

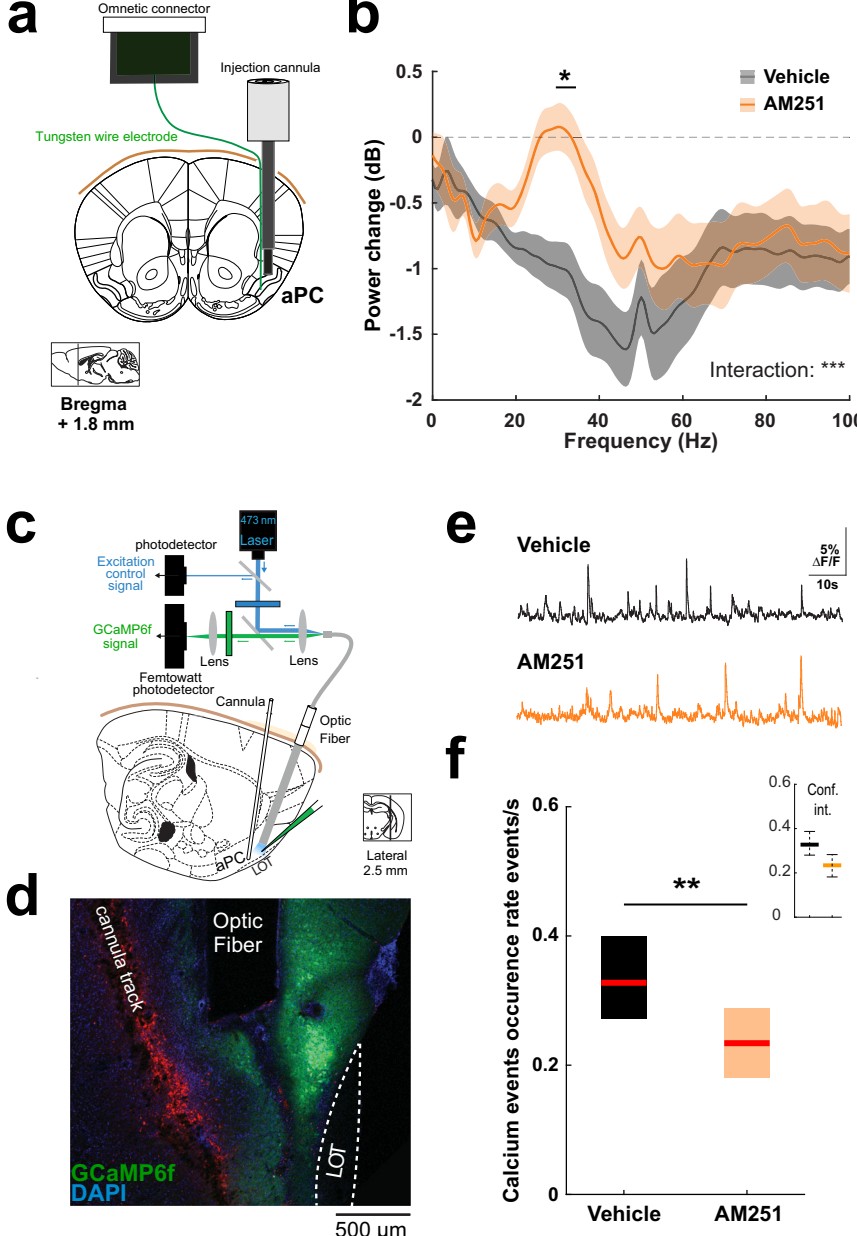

**Fig. 3 | Effect of local CB1 receptor blockade on gamma oscillations and spontaneous calcium events in the anterior piriform cortex. a** Schematic of the implanted device: a tungsten wire electrode is attached to a local injection cannula for LFP monitoring in the vicinity of local drug infusion in freely moving mice. Adaptation from Paxinos and Watson mouse brain atlas. **b** Mean change in power spectrum induced by Vehicle (black) or AM251 (orange) local injections. Means, bold lines; SEM, shaded areas. Two-way ANOVA interaction: ***$P < 0.001$. Horizontal line indicates post-hoc $t$ tests with Bonferroni correction: *$P < 0.05$. $n = 14$ sessions from 8 mice for both Vehicle and AM251. **c** Schematic of the experimental approach: fiber photometry system to record anterior piriform cortex (aPC) excitatory cells calcium signal (GCaMP6f is expressed under CamKII promoter)

following local drug delivery in vivo. Adaptation from Paxinos and Watson mouse brain atlas. **d** Histological sagittal section showing the recording/injection site. Blue, DAPI; green, GCaMP6f; red, autofluorescence with position of injection cannula. Similar histological verifications were conducted in all recorded mice. **e** Example of spontaneous calcium events monitored following local injection of Vehicle and AM251. **f** Median occurrence rate of spontaneous calcium events. Vehicle 0.33 [0.27 0.40] events/s vs AM251 0.23 [0.18 0.29] events/s. Two-sided Mann–Whitney test: **$P = 0.002$; $n = 16$ and 20 hemispheres from 8 and 10 mice for Vehicle and AM251, respectively. Boxplots and values in the legend represent median and 25–75th percentiles ([lower bound upper bound]). Source data are provided as a Source Data file.

mainly through PopEvents (Fig. 1g, h), we examined the relationship between the changes in gamma oscillation amplitude and the changes in PopEvents occurrence rate across sessions. We found a negative correlation between the changes in gamma amplitude and PopEvent rate (Supplementary Fig. 4c). Altogether, these observations show that gamma activity and neuronal synchrony in PopEvents are linked in the aPC, suggesting that these two functions are regulated by a common mechanism involving endocannabinoid signaling.

## Local injections of CB1R antagonists impair population calcium transients and increase gamma oscillations in the aPC

To test whether local CB1R blockade increases gamma oscillations in freely moving mice, we coupled a tungsten-wire electrode to injection cannulas chronically implanted in the aPC to record LFP signals in the immediate vicinity (~500 μm) of the sites of local antagonist delivery[37] (Fig. 3a, Supplementary Fig. 5a). We observed that Vehicle local injections consistently induced a power drop in the LFP signal,

probably due to the slight hydrophobic nature of the solution as well as infusion-induced tissue distortion. Yet, in line with the results obtained with systemic blockade of CB1R, local infusions of a CB1R antagonist (AM251, 4 µg, 10–15 min before recording starts) significantly increased low gamma frequency power when compared to Vehicle injections (Fig. 3b). In contrast, when we injected tetrodotoxin (TTX, 5 µg) through the same cannulas at the end of our experiments, we observed a strong reduction of LFP power in all frequency bands (Supplementary Fig. 5b), highlighting the specificity of the effect on low gamma obtained with the antagonist. These data indicate that the endocannabinoid system actively down-regulates gamma oscillations within the aPC. What about the effect of local CB1R blockade on neuronal synchronization? Since the detection of PopEvents relies on unit activity and cannot be achieved with LFP signals only, we opted for a fiber photometry approach that allows detecting population calcium events[52] and indirectly studying neuronal activity. We recorded population activity in aPC neurons expressing the calcium reporter GCamp6f in excitatory neurons of the aPC (with the CaMKII promoter) while performing local drug infusion[53] to block CB1Rs in the aPC of freely moving mice (Fig. 3c, d). Upon excitation with low laser intensity (0.05–0.1 mW), the bulk calcium signals were collected using an optic fiber implanted in layer II/III where GCamp6f was expressed (Fig. 3d) and the emitted fluorescence was continuously acquired. In awake, freely moving mice, aPC imaging showed spontaneous regular stereotyped positive fluorescence transients with sharp onsets (Fig. 3e), reminiscent of calcium signals observed during synchronous population events[52]. We focused on these spontaneous calcium events and observed that they exhibited a similar occurrence rate as compared to the PopEvents we recorded with electrophysiology (PopEvents in Vehicle: median 0.39, 95% confidence intervals (CI) [0.31 0.52] vs Calcium transients in Vehicle: median 0.33, CI [0.28 0.39]; Figs. 1e and 3e, f). Following acute local infusion of the CB1R antagonist (AM251, 4 µg) in the vicinity of the imaging site, we observed a strong decrease in the occurrence rate of calcium transients (Fig. 3e, f). These data suggest that CB1Rs are physiologically active within the aPC where they favor the emergence of presumed bursts of activity in the local network.

## CB1R blockade impairs neuronal synchrony during odor presentations

So far, our results indicate that CB1R blockade leads to an impaired synchronization of aPC networks in baseline conditions, which could impact animals' ability to detect incoming stimuli. We next wondered whether the desynchronization induced by CB1R antagonists would be maintained upon olfactory stimulations. To address this question, we locally infused CB1R antagonist before conducting fiber photometry experiments in the presence of odorants (benzaldehyde or isoamyl-acetate, 10% −3 successive presentations for each odorant) diluted in mineral oil and introduced in the ventilated recording chamber through an olfactometer (Fig. 4a, b). Neuronal calcium responses were recorded and normalized to pre-odor epochs (Fig. 4c; Methods). Consistent with sensory adaptation[54], neuronal responses decreased across the 3 odor presentation trials, regardless of the odorant identity and the solution infused (Supplementary Fig. 6). As compared to control Vehicle infusions, we found that AM251 strongly blunted aPC odor-evoked population calcium responses (Fig. 4c, Supplementary Fig. 6a), independently of the nature of the odorant (Supplementary Fig. 6c), while calcium signal in control "odorless" (mineral oil) trials was not affected by AM251 infusions. The AM251-induced reduction in the fiber photometry signal observed under stimulus presentation could be explained either by a strong decrease in neuronal activity or by a decoupling of neuronal firing among the population of odor-activated neurons.

In order to test for these possibilities, we performed unit silicon probe recordings in head-fixed mice during controlled odor presentations (Fig. 4d, e) upon blockade of CB1R signaling or in control

conditions. Benzaldehyde and isoamyl-acetate were randomly presented over the course of 30 trials at 3 different concentrations using an olfactometer. Respiration was monitored with pressure sensors connected to an intra-nasal cannula[55], allowing the alignment of neuronal responses to inhalation onset in odor or in baseline conditions[15] (Methods, Fig. 4e). We found that Rim injections increased the evoked firing rates to the odorant but not to the control (mineral oil) presentations (Fig. 4f). This effect was due to a decrease in baseline firing rates after Rim injection, with odor-evoked firing rates maintained (Supplementary Fig. 7). The conservation of the odor-evoked spiking indicates that CB1R blockade in aPC does not reduce neuronal activity during odor presentation. To assess the possibility of a decoupling of neuronal firing, we computed odor-evoked signal correlations for pairs of recorded units and found that it was reduced in the presence of Rim as compared to Vehicle injections (Fig. 4g; Supplementary Fig. 8). Noise correlations were also lowered (Supplementary Fig. 9) whereas gamma oscillations were generally potentiated in Rim (Supplementary Fig. 10) as observed in freely moving mice (Fig. 2b). These results indicate that CB1R blockade reduces the coordination among neuronal populations not only in baseline conditions, but also during odor presentations.

## Local injection of CB1 receptor antagonist enhances olfactory perception

In other sensory systems, the neocortex has been shown to alternate between synchronized and desynchronized states, with desynchronization being linked to a "perceptive" state associated with reliable sensory responses[1–7]. On the other hand, gamma oscillations are related to olfactory perception in rodents[48,50] and in the human olfactory cortex[56] and they have been linked to sensory processing in other sensory systems[10,57]. Since our findings indicate that CB1R blockade impairs neuronal synchrony and increases gamma oscillations in the aPC, we reasoned that selective blockade of CB1R in the aPC could favor the perceptive state of animals and facilitate olfactory perception. To test this hypothesis, mice were bilaterally implanted with guide cannulas and we locally injected the CB1R antagonist or its Vehicle before conducting an olfactory detection task (Fig. 5a, b; Supplementary Fig. 11a). In this task, water-restricted mice were sequentially presented with increasing concentrations of neutral odors (from 0.001% to 1%), and the time they spent investigating the odor source was measured[33]. The first odor concentration that was actively explored by a mouse (as compared to oil "no odor" control) was considered as a likely measure of the olfactory detection threshold of that individual[33] (see Methods). Regardless of the odor used (Supplementary Fig. 11b, c), animals displayed different exploratory behaviors following Vehicle injections as compared to AM251 injections. Upon Vehicle injections, the detection thresholds were dispersed across individuals, with most mice displaying thresholds at a concentration of 0.01% or 0.1% (14/23 mice; Fig. 5b, c; Supplementary Fig. 11b, c, f). Conversely, most mice treated with AM251 (9/11 mice) had detection thresholds at the lowest concentration tested, 0.001% (Fig. 5b, c; Supplementary Fig. 11f). Importantly, these pharmacological manipulations did not affect the total exploration time of the tested mice (Supplementary Fig. 11d, e) and did not alter odor habituation (see below and Supplementary Fig. 11h, i).

Next, we tested whether similar results could be obtained from another behavioral paradigm. Mice were water-restricted and habituated to having access to two bottles of water for 1 h every day. On the day of the test, animals had the choice between one bottle of water or a bottle containing a solution containing a novel odor. Because rodents display spontaneous neophobic avoidance when perceiving new odorized solutions, they normally prefer to drink water over the scented solution[58,59]. As expected, mice treated with local injections of the CB1R antagonist AM251 or its Vehicle showed avoidance of the new odorized solution, indicating that they were able to perceive the odor

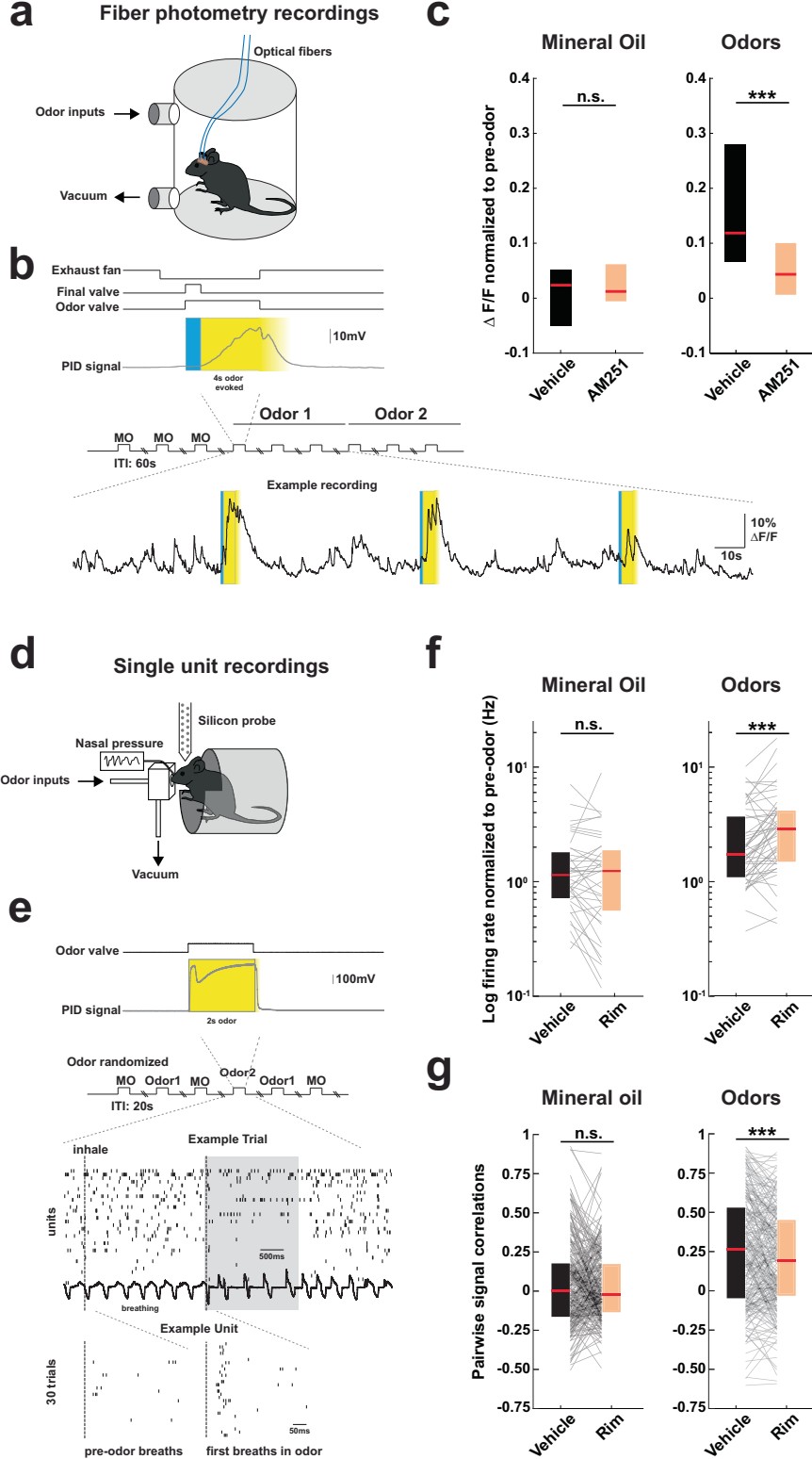

at the first concentration tested (Fig. 5d). When the odor was diluted 100 times, Vehicle-injected animals no longer showed avoidance towards any solution whereas AM251-treated mice avoided the odorized solution, indicating that they were still able to perceive the odor despite the dilution (Fig. 5e). Importantly, the total liquid consumption of mice was not altered (Supplementary Fig. 11g), indicating that the motivation to drink was not affected by the treatment. Altogether, these results suggest that endocannabinoid signaling in the aPC

modulates olfactory detection, thereby preventing behavioral responses to low odorant concentrations.

Finally, we wondered whether endocannabinoids in the aPC could affect other olfactory abilities. In a previous study, we demonstrated— adopting similar injections to target the aPC—that CB1R blockade impairs retrieval of appetitive olfactory memory[31]. Using a similar behavioral assay as for odor detection, we investigated the impact of CB1R in the aPC in olfactory discrimination with a habituation-

**Fig. 4 | Effect of CB1 receptor blockade on odor-evoked responses. a** Illustration for odor presentation coupled with fiber photometry in freely moving mice. **b** Top, schematic of experimental design. Bottom: example of calcium signal following odor presentation. **c** Median calcium signals recorded with mineral oil or odors following Vehicle (black) or AM251 (orange) local anterior piriform cortex (aPC) injections. Mineral oil (MO): Vehicle 0.024 [0.050 0.051]; AM251 0.013 [−0.005 0.062] (*P* = 0.564). *n* = 14 and 20 hemispheres from 8 mice and 10 mice for Vehicle and AM251, respectively. Odors: Vehicle 0.119 [0.066 0.279]; AM251 0.044 [0.007 0.100] (\*\*\**P* = 9.17e−4). *n* = 28 and *n* = 40 recordings from *n* = 14 and 20 hemispheres for each odor, from 8 mice and 10 mice for Vehicle and AM251, respectively. **d** Illustration for odor presentation coupled with in vivo head-fixed single unit and respiration recordings. **e** Top, a schematic of the experimental design. Middle: example of raster plots from E_Cells and respiration in the presence of odor (gray shaded area). Bottom: example of raster plot of a unit aligned on first breaths

in odor. Vertical dashed line: inhalation onset. **f** Median normalized firing rate of E_Cells (see Methods) following Vehicle or Rim systemic injections. Mineral oil: Vehicle 1.143 [0.732 1.768]; Rim 1.238 [0.570 1.840] Hz (*P* = 0.22). *n* = 40 odor-responsive cells. Odors: Vehicle 1.727 [1.116 3.623]; Rim 2.869 [1.525 4.079] Hz (\*\*\**P* = 1.65e−04). *n* = 54 odor-responsive cells (14 responded to both odors and appeared twice). **g** Median pairwise signal correlations of E_Cells (see Methods) following Vehicle or Rim injections. Mineral oil: Vehicle 0.004 [−0.158 0.170]; Rim −0.022 [−0.128 0.166] (*P* = 0.66). *n* = 279 odor-responsive cell pairs. Odors: Vehicle 0.264 [−0.041 0.524]; Rim 0.195 [−0.025 0.445] (\*\*\**P* = 4.11e−04). *n* = 254 odor-responsive cell pairs. Olfactometer data were collected in 3 sessions from 2 mice. Two-sided Mann−Whitney tests were used to assess significance. Boxplots and values in the legend represent median and 25−75th percentiles ([lower bound upper bound]). n.s. non-significant. Source data are provided as a Source Data file.

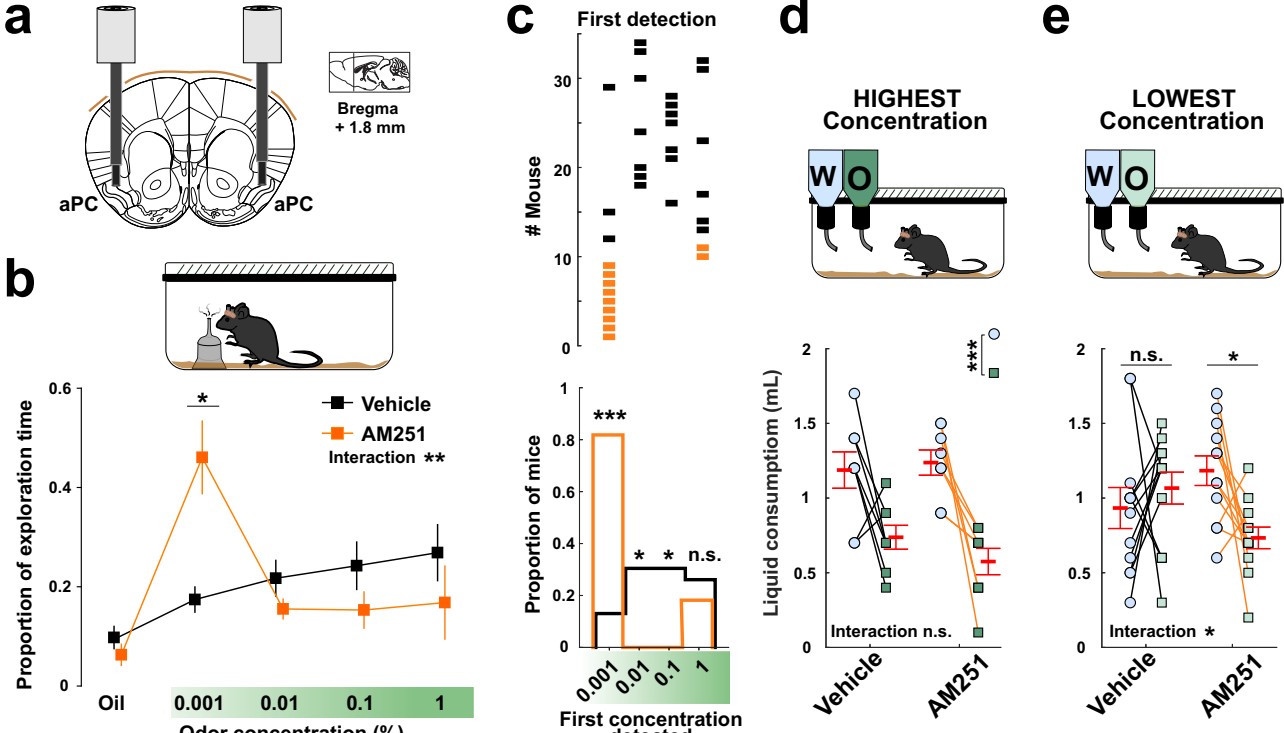

**Fig. 5 | Effect of local injections of CB1 receptor antagonist on olfactory detection. a** Schematic of bilateral injection cannulas implanted into the anterior piriform cortex (aPC). Adaptation from Paxinos and Watson mouse brain atlas. **b** Top: illustration of explorative odor detection task (see Methods). Bottom: proportion of exploration time for mineral oil and increasing odor concentrations. Mean, square symbols; SEM, outer lines. Two-way ANOVA interaction: \*\**P* = 0.003. At 0.001% (mean ± SEM): Vehicle 0.17 ± 0.027 vs AM251 0.46 ± 0.075; \**P* = 0.02 (post-hoc *t* test with Bonferroni correction). **c** Top: distribution of individual mice according to the concentration they first detected following Vehicle (black) or AM251 (orange) local injections. Bottom: proportions of mice with first detection at specified concentration. Two-sided Chi square tests: at 0.001%, 0.13 in Vehicle vs 0.82 in AM251 (\*\*\**P* = 8.64e−05); at 0.01%, 0.30 vs 0.00 (\**P* = 0.04); at 0.1%, 0.30 vs 0.00 (\**P* = 0.04); at 1%, 0.26 vs 0.18 (*P* = 0.571). *n* = 23 and 11 mice for Vehicle and

AM251, respectively. **d** Top: illustration of the spontaneous odor aversion task (see Methods). A high concentration of a new odorized solution is compared to pure water. Bottom: total consumption of the two solutions. W, water (light blue); O, odor (dark green). Vehicle: water (mean ± SEM) 1.19 ± 0.12 mL vs odor 0.74 ± 0.08 mL. AM251: water 1.24 ± 0.08 mL vs odor 0.58 ± 0.09 mL. Two-way ANOVA: interaction *P* = 0.38; solution effect \*\**P* = 0.003. *n* = 8 for both Vehicle and AM251. **e** Same as (**d**) but with odorized solution diluted 100 times as compared to (**d**), represented in light green. Vehicle: water (mean ± SEM) 0.93 ± 0.14 mL vs odor 1.07 ± 0.11 mL. AM251: water 1.18 ± 0.10 mL vs odor 0.73 ± 0.07 mL. Two-way ANOVA: interaction \**P* = 0.04. Post-hoc *t*-test with Bonferroni correction: Vehicle, water vs odor, *P* = 0.97; AM251, water vs odor, \**P* = 0.05. *n* = 12 mice for both Vehicle and AM251. Red symbols, mean ± SEM. n.s., non-significant. Source data are provided as a Source Data file.

dishabituation task[33]. In this task, animals were exposed to 2 sequential presentations of the same odor (Odor 1—either isoamyl-acetate or benzaldehyde)—followed by the presentation of a different odor (Odor 2; Supplementary Fig. 11h). Because mice tend to explore more odor that they consider as novel, discrimination performance can be measured by comparing the time the animal spends exploring the familiar odor (Odor 1) as compared to the second novel odor (Odor 2). We chose odor concentrations based on the results obtained in the

odor detection task to circumvent the fact that detection thresholds are different in Vehicle and AM251 locally injected mice (Methods). Both groups of mice showed a decreased exploration when Odor 1 was presented for the second time (Odor1'−Supplementary Fig. 11i), indicating that CB1R activation in the aPC is not required for habituation to a given olfactory stimulus. Upon the presentation of the novel stimulus, however, Vehicle-treated mice showed increased exploration of the novel odor, showing that they discriminated this new odor from

the one presented before. In contrast, AM251-treated mice did not show any regain of exploration relative to the familiar odor (Supplementary Fig. 11j). This experiment suggests that blockade of CB1R in the aPC impairs the discrimination of distinct olfactory stimuli. Altogether, these data suggest that endogenous CB1R signaling in the aPC plays a role in the trade-off between odor detection and discrimination.

## Discussion

In this study, we examined the functional role of CB1Rs in olfactory processing and the internal circuit dynamics of the anterior piriform cortex using silicon probe recordings, fiber photometry and pharmacological manipulations in freely behaving mice. We found that systemic CB1R blockade decreased the occurrence of spontaneous PopEvents and reduced neuronal co-firing among piriform units, specifically during these events. Surprisingly, this effect was inversely correlated to a prominent increase in gamma oscillations. Similar observations were obtained after a local blockade of CB1Rs within the aPC that reduced the occurrence of population calcium transients measured with fiber photometry while increasing low gamma oscillation power. Increased gamma power and reduced signal correlation were also observed during odor presentations following local and systemic CB1R blockade, indicating that CB1Rs generally regulate neuronal coordination. Using three distinct behavioral tasks, we found that mice have lower odor detection thresholds and altered odor discrimination when CB1Rs are pharmacologically blocked in the aPC. These results are consistent with the idea that endogenous cannabinoids favor population co-firing in the aPC at the expense of gamma oscillations, resulting in reduced abilities to detect incoming olfactory stimuli.

### Endogenous activation of CB1R in vivo tunes spike timing coordination in the aPC

In line with preceding reports[44–46], our extracellular unit recordings and fiber photometry experiments highlighted the presence of PopEvents and population calcium transients in the aPC. Interestingly, these population events are reduced following systemic or local aPC blockade of CB1Rs, indicating a role of CB1Rs in orchestrating spike timing synchrony during PopEvents and showing that these receptors are endogenously activated in vivo. This result is in line with previous studies showing that exogenous[37–39] or endogenous[36] activation of CB1Rs has an impact on neuronal coordination in other brain regions. Knocking out CB1Rs in the secondary (but not in the primary) visual cortex has been shown to increase correlated calcium transients[36]. The present data suggest that endogenous activity of CB1R favors neuronal co-activation, specifically during PopEvents in aPC circuits. In contrast, injections of a CB1R antagonist were not associated with any significant physiological changes in the hippocampus[37], suggesting little endo-cannabinoid control of neuronal coordination in that brain area. Altogether, these observations highlight the region-specific effects of endocannabinoid signaling. In the piriform, it was proposed that PopEvents result from the recurrent excitation within neuronal circuits[45,46]. Knowing that CB1Rs are primarily expressed by inhibitory interneurons in this region[31] and that CB1R activation usually suppresses neuronal transmission[26], it seems plausible that CB1R antagonist administration results in an increased inhibitory transmission in the aPC. In turn, this increased inhibition would prevent the emergence of synchronous PopEvents in the recurrent network. This is consistent with previous work showing that suppressing inhibition in aPC decreases neuronal coordination with regard to respiration[15]. Altogether, our work supports the idea that the physiological activity of CB1R plays a role in the fine temporal coordination of intact neuronal circuits in vivo. It remains to be addressed whether endogenous CB1R activity similarly regulates network activity in other sensory neocortices, such as the somatosensory cortex, where neuronal

coordination has been observed along with lower response reliability to external inputs[1–7].

### Gamma and spike timing synchrony in the aPC

The very first description of gamma oscillations in the brain comes from the seminal recordings of Lord Adrian in the olfactory bulb of mammals[51]. While gamma rhythms are now considered ubiquitous across brain areas and across species[57,60], they remain a hallmark of olfactory circuits[48–50,61], including the piriform cortex[62–65]. Our results show that systemic injections of the CB1R antagonist Rim induce a robust increase in gamma power as well as increased entrainment of both E and I_Cells to gamma oscillations (Fig. 2 and Supplementary Fig. 3). The increase in gamma power induced by Rim administration was correlated with the reduction in spike timing synchrony between pairs of neurons (Supplementary Fig. 4). These observations suggest that gamma oscillations and short-time scale neuronal coordination in the aPC are linked by a common mechanism. Consistent with previous works in the aPC[64] or in other brain regions[66], our results suggest that gamma oscillations partly depend on mutual interactions between E and I neurons within the aPC as both cell types are phase-locked to the oscillation and increase their modulation when CB1R is blocked. Yet, aPC gamma also relies on external inputs from the olfactory bulb (OB), a documented source of gamma oscillations[50]: indeed, our CSD analysis revealed current sinks at the level of the superficial layer of the aPC, precisely where afferences from the OB are located (Supplementary Fig. 3a). These observations are in line with reports showing that suppression of OB afferences to the aPC eliminates gamma oscillations in the piriform[12,65] and that OB stimulations trigger gamma in the aPC[64]. On the other hand, the mechanism underlying PopEvents likely relies on the recurrent excitation among piriform E_Cells, as proposed in earlier studies[45,46]. The relative contribution of internal (i.e., from the recurrent network) and external (i.e., OB-related) inputs to aPC function—that both act in a synergistic manner even at the single cell level[67]—could, therefore, be influenced by the ECS in a way reminiscent of the reported impact of acetylcholine[23]. Based on the expression pattern of CB1Rs[31], we suspect that CB1R-expressing inhibitory neurons play a central role in this modulation.

### Impact of CB1R signaling on olfactory stimulus responses

Our results show that CB1Rs do not only impact neuronal coordination in baseline conditions, but they also do so when odor stimuli are presented. Precisely, we found that local blockade of CB1Rs in the aPC reduced population calcium responses of E_Cells to odors when these were recorded with fiber photometry. These decreased responses were most likely explained by a decorrelation among neuronal populations, as unit recordings during CB1Rs blockade showed a decreased signal correlation when stimuli were presented via an olfactometer (i.e., responses of pairs of neurons were more dissimilar with regards to each other) (Fig. 4). Although signal correlation and PopEvents are independent phenomena (being related to odor presentations or baseline conditions, respectively), they are both influenced by the degree of co-activation within neuronal populations.

Like others before, we suspect that the strong recurrent connections among aPC units support correlated activity central to pattern completion and, therefore, memory processes[46] when a stimulus is presented. In contrast, the decorrelation of aPC neurons in response to a given olfactory stimulus—such as the one we observed here—could increase the coding capacity of the networks by increasing the dimensionality of neuronal representations. In parallel, the reduced noise correlation and increased evoked responses we observed during repeated stimulus presentations in head-fixed mice could increase the reliability and the efficiency of neuronal responses to odorants. Interestingly, in the olfactory bulb, optogenetically-evoked synchronous spikes are more likely to be detected than asynchronous spikes[68]. Although the effect could be different in the aPC, one could imagine

that synchronous "relevant" spikes would be better detected if non-relevant spikes are decorrelated. CB1Rs activity in the aPC could play a key role in modulating this "background" level of synchrony before a stimulus comes in, thereby tuning the signal-to-noise ratio based on the fine temporal properties of neuronal firing. Indeed, we believe that neuronal responses to olfactory stimuli have to be considered in combination with the ongoing network activity that can influence the detectability of the incoming sensory inputs[1–11]. Our study provides evidence that CB1Rs control the state of this ongoing internally-generated activity.

## Impact of CB1R signaling on internal network state

Accumulating evidence suggests that internal brain states are key determinants of sensory perception[1–7]. Brain states relate to the degree to which neuronal activity in the cerebral cortex is synchronized. For instance, running or whisking triggers a desynchronized state where neuronal coordination is decreased in the mouse barrel cortex, resulting in sensory responses that are more reliable and show better signal-to-noise ratios[1,7]. In the piriform cortex, single-unit[69,70] and LFP[17,44] responses are attenuated during slow wave state as compared to waking. Although the most striking changes in brain states are seen between waking and sleep, more subtle variations in neuronal synchronization are also seen within waking, with increased or reduced synchronization between neurons during quiet wakefulness or whisking, respectively[11]. The physiological roles of endocannabinoid signaling in this phenomenon—and more generally in sensory perception—remain largely unexplored.

Our study shows that CB1R blockade in the aPC is associated with a decreased occurrence of calcium transients in this area, consistent with lower neuronal coordination (Fig. 3f). In line with the fact that neuronal desynchronized states favor sensory responses[10,11], we found that local CB1R blockade in aPC lowers odor detection thresholds, making mice more sensitive to low concentrations of a novel odor (Fig. 5). The endogenous activation of CB1Rs may therefore tune cortical internal dynamics by promoting spiking synchrony, thereby contributing to lower detection of external stimuli. As a complementary mechanism, the reduction of gamma oscillations by the endogenous activation of CB1Rs could participate in reducing olfactory detection abilities in physiological conditions.

## CB1Rs activation controls olfactory behaviors

Our behavioral results may seem at odds with previous work reporting that CB1R *activation* in the main olfactory bulb decreases cortical feedback and enhances odor detection[33]. However, CB1Rs are expressed differently in the piriform cortex and in the bulb: in piriform, the density of CB1Rs is higher at inhibitory synapses than excitatory ones[31], which is the opposite of the bulb[33]. These differences in the patterns of expression likely produce different impacts on neuronal networks. Moreover, mice were food-deprived in the previous study while it was not the case in the present study. Since fasting has been shown to change cannabinoid signaling[71], this parameter can contribute to differences with regard to odor detection performances.

While aPC neuronal synchrony—favored by CB1R activation—coincides with reduced olfactory detection abilities, it seems to be the opposite when it comes to discrimination and memory performances. In a previous study, we found that pharmacological blockade or genetic deletion of CB1Rs in the aPC impairs the retrieval of appetitive olfactory memory[31]. Our current results show that blocking CB1Rs reduces PopEvents occurrence rate and impairs olfactory discrimination (Supplementary Fig. 11j). These observations suggest that synchronous population bursts could facilitate olfactory memory[31] and discrimination processes. In fact, the PopEvents, presumably supported by recurrent connections among aPC excitatory cells[44], have previously been described during slow-wave sleep[44,45]. These events are highly reminiscent of neuronal activity observed during sharp-wave ripples[72] (SWRs) in the hippocampus, another paleocortical area: SWRs are also observed during rest and slow-wave sleep and largely depends on recurrent excitatory connections among CA3 hippocampal neurons[72]. SWRs are thought to play a key role in memory consolidation and memory retrieval as they orchestrate the reactivations of past activity patterns[72]. Further work will be required to investigate whether PopEvents in the aPC play similar roles and whether the alteration in discrimination performances when CB1R are blocked in the aPC, is linked to a modulation of the associative recurrent circuit or to other mechanisms.

To conclude, this study reveals a role for the ECS in the physiological regulation of sensory processes. In the aPC of awake, freely behaving mice, CB1R blockade modifies circuit properties known as related to perception: gamma oscillations and short-timescale spike coordination. Consistent with the role of these network features in sensory responses, the blockade of CB1Rs in the aPC favors the detection of low concentrations of odorants. We propose that, in the aPC, the endogenous activation of CB1Rs promotes internal network synchrony at the expense of gamma oscillations, thereby tuning olfactory perception.

# Methods

## Animals

All experimental procedures were approved by the local Committee on Animal Health and Care of Bordeaux (CEEA 50), the local Committee on Animal Health and Care of Institut Pasteur, the French Ministry of Agriculture and Forestry (facility authorization number A33063098, A33063940, and A7515019) and the Ministry of Education and Research. Adult (2–5 month-old) male mice born in captivity were used in this study (i.e., only male mice were used in experimental study design and analysis). Mice carrying the "floxed" CB1 gene [CB1 f/f - CB1-flox] were used for behavioral testing with cannula (a genotype consistent with our previous studies[31,32]), and C57BL/6 J mice were used for in vivo electrophysiology and fiber photometry imaging. All behavioral experiments were performed during the light phase. Animals were kept in individual cages after surgery under a 12 h light/dark cycle and were maintained under standard conditions with food and water *ad libitum* (temperature and hygrometry within the following ranges: 20–24 °C and 50–70, respectively). Access to drinking water was controlled in the case of specific behavioral procedures (see below). All animals were handled and habituated to receiving local or systemic injections prior to the reported experiments.

## Surgeries

**Stereotaxic Viral injections and fiber implantation for fiber photometry experiments.** Mice were anesthetized by intraperitoneal (i.p.) injection of a mix of ketamine, (100 mg/kg), xylazine (10 mg/kg) and buprenorphine (0.05 mg/kg) and positioned in a stereotaxic frame. After local anesthesia (lidocaine) followed by skin incision and skull craniotomy, mice were injected bilaterally using pulled glass capillaries connected to a Nanoject System (Drummond) in the aPC (AP, +1.6 mm; ML, ±2.5 mm; DV, −4 mm from brain surface; 250 nL in 4 min) with GCaMP6f-expressing viral vector (GCamp6f construct was provided by the GENIE Project, Janelia Farm Research Campus, Howard Hughes Medical Institute; the AAV.CamKII.GCaMP6f.WPRE viral vector, Addgene plasmid # 100834, was a gift from James M. Wilson and produced by Upenn Vector Core; 3.10E + 13 viral genome/mL). Following viral injection, an optic fiber (multimode, 425 μm diameter, NA 0.50, LC zirconia ferrule) associated with a stainless-steel guide-cannula was implanted bilaterally above the virus injection site (AP: +1.6; ML, 2.5; DV, −3.9 mm from brain surface) and stabilized with acrylic and dental cement. The stainless steel guide cannula (26 gauge, 7 mm long) was positioned ~2 mm aside from the fiber and 4 mm above the tip of the fiber with a ~25° angle so that the tip of the injection cannula was close to the imaging field (see Fig. 3c, d). Mice were then

returned to their home cage, monitored daily, and left to recover for 3 weeks after injection. Postsurgical analgesia (0.05 mg/kg buprenorphine) was provided via subcutaneous injection over the 48-h period post-surgery.

**Guide cannula implantation for behavioral tasks and local field potential recordings.** Animals were anesthetized with i.p. injections of a mix of ketamine (100 mg/kg) and xylazine (10 mg/kg) or with isoflurane (1.5%). After local anesthesia (lidocaine) followed by skin incision and skull craniotomy, a bilateral 3.5 mm long stainless steel guide cannula (Bilaney, UK) was implanted in order to target the aPC[31] (AP, +1.6; ML, ±2.5; DV, 4.5 mm from the skull). For LFP recordings coupled with local pharmacology experiments (Fig. 3a, b and Supplementary Fig. 4), tungsten wires (50 μm, California Fine Wires) were attached 1 mm below the tip of the guide cannulas. Since injectors protruded 0.5 mm from these tips, tungsten wires recorded an LFP signal of -0.5 mm from injection sites. Guide cannulas were secured on the skull surface with dental cement. Stereotaxic coordinates and drug diffusion were verified *post-hoc* via histology and with 1,1′-dioctadecyl-3,3,3′3′-tetramethylindocarbocyanine perchlorate (DiI, Sigma Aldrich) injections (stock solution: 2.5 mg DiI diluted in 1 mL DMSO; stock solution diluted 10% in saline before injections): like AM251, DiI is a hydrophobic molecule that might approximately mimic the drug diffusion spread. Postsurgical analgesia (0.1 mg/kg buprenorphine) was provided via an injection of buprenorphine (0.1 mg/kg, subcutaneous) or meloxicam (5 mg/kg, i.p.) and mice were allowed to recover for 1–2 weeks before the beginning of the experiments. The placement of aPC cannula was verified using an injection of 2% pontamine sky blue solution (0.5 μL per side[31]) 10 min before animal sacrifice.

**Implantations of silicon probes for in vivo electrophysiology.** Seven out of nine mice were implanted with a probe in the hippocampus CA1 and another one in the aPC. The two other mice were implanted only in the aPC. In 5 of the mice, respiratory rates could also be recorded via a nasal cannula implant[55]. For electrode implantation surgeries, mice were anesthetized with isoflurane (1.5%) and received an i.p. injection of meloxicam (5 mg/kg). After local anesthesia (lidocaine) followed by skin incision, craniotomies were performed under stereotaxic guidance. Silicon probes (Buzsaki32 Neuronexus probes with 32 channels for the hippocampus; Buszaki64 Neuronexus or H6 Cambridge Neurotech probes with 64 channels for the aPC) were mounted on custom-made 3D-printed micro-drives to allow precise adjustment of electrodes' positions after implantation. The two probes (targeting CA1 and aPC) were inserted ipsilaterally to each other, above the target region (AP, −1.8; ML, ±1.4; DV, 0.7 mm from brain surface for CA1 and AP, +2; ML, ±2.2; DV, 2.5 mm for the aPC). Craniotomies were sealed with silicon (3–4680; Dow Corning). A stainless steel screw placed above the cerebellum was used as a ground and reference electrode. Finally, a copper mesh was attached to the skull with dental cement and connected to the ground screw to protect the probes and act as a Faraday cage. Animals were allowed to recover for at least one week with *ad libitum* food and water before starting the experiment. During post-surgery recovery, probes were moved gradually until the desired position was reached. Hippocampal and aPC cellular layers were identified physiologically by the presence of sharp wave ripple oscillations[72] (for CA1) and the absence of up/down states together with population event activity[44,45] and pronounced gamma oscillations (for aPC). Recording locations were systematically confirmed by electrolytic lesions (10 μA for 10 s) and histology (see Fig. 1a). In order to maximize the number of sessions and the use of mice, three recording sessions with Vehicle and two sessions with Rimonabant were obtained from water restricted animals (no difference was observed as compared to non-restricted animals).

## Experimental design and data acquisition

**Drugs.** To study the ECS, CB1Rs were blocked using inverse agonists/antagonists (local aPC injections: AM251 from Tocris Bioscience; systemic injections: Rimonabant from Cayman Chemical). The concentrations used were chosen based on previous works both for systemic (1 mg/kg of Rimonabant[31,73–76]) and local injections (AM251 4 μg[33,77,78]) of CB1R antagonists.

AM251 was dissolved in a mixture of warm 10% Cremophor-EL, 10% DMSO and 80% saline (NaCl 0.9%), prepared immediately before injection. The vehicle solution was a mixture of 10% Cremophor-EL, 10% DMSO, and 80% saline (NaCl 0.9%). AM251 (4 μg/0.5 μL per side) or its Vehicle was injected bilaterally in the aPC using silicone tubing connected to a peristaltic pump (PHD 22/2000 Syringe Pump Infusion, Harvard Apparatus). For local manipulations coupled to LFP recordings, injections were performed 10 min before starting the recordings to limit the effects of stress. To verify the effect of local pharmacological manipulations on LFP signals, tetrodotoxin citrate (TTX, 5 μg/0.5 μL per side, Tocris Bioscience) was injected 10 min prior to recordings. The injections were performed 10–25 min before starting the odor detection and fiber photometry experiments.

Rimonabant was dissolved in a mixture of 1.25% Tween 80, 1.25% DMSO, and 97.5% saline (NaCl 0.9%). The vehicle solution was a mixture of 1.25% Tween 80, 1.25% DMSO, and 97.5% saline (NaCl 0.9%). After 1 h of baseline recordings, animals were injected with Rimonabant (i.p.; 1 mg/kg) or its Vehicle or with Vehicle and 2 h later with Rimonabant.

**Explorative odor detection task.** The explorative odor detection task was adapted from previous studies[33]. Briefly, the materials used consisted of a test cage similar to the home cage but with an odor holder that was made from the stainless-steel lid of an unused mouse water bottle. The bottle lid fits flush in a hole, which prevents it from being displaced by the mice during the test. The tip of the lid's nozzle extended 5 cm and contained a 3 mm hole from which odors could emanate. For each mouse, 2–3 mm of fresh sawdust was mixed with the mouse's home cage sawdust. A camera was placed above the test cage and a hole was cut in the cage filter paper lid to allow easier visualization of the odor holder during behavior. The cage was placed on a table in a dedicated testing room (separate from the housing room) at 70 ± 5 lux.

The behavioral protocol consisted of 3 days of habituation to the test cage before the test day. Each mouse was placed in the test cage for 3 min each of 5 separate trials for the odor detection experiment with a 3-min inter-trial interval. Before each trial, 10 μL of mineral oil was placed on a 2 cm strip of filter paper and placed in the odor holder. To habituate the mice implanted with a cannula to be injected, saline injection was performed on the 3rd day of habituation with a nanoliter injector. All mice were water-deprived after the 3rd day of habituation for 24 h before the test to increase their motivation to explore. Local injections into the aPC were performed 10 min before the first trial at the infusion rate of 0.5 μL per minute for a total volume of 0.5 μL infused per side.

Iso-amyl acetate [banana-like (Sigma-Aldrich)] and benzaldehyde [almond-like (Sigma-Aldrich)] odors were presented in a counterbalanced schedule across all experiments. Based on a serial dilution method with mineral oil, increasing concentrations of the odors were tested as follows: mineral oil, 0.001%, 0.01%, 0.1%, and 1%. 10 μL of these solutions were placed on a strip of filter paper immediately before each trial and then introduced inside the nozzle of the odor holders. The odor holders were cleaned with 30% ethanol between trials and experiments.

The time mice spent investigating the presented odor was counted manually using a customized program (BehavScor v3.0 beta): considered epochs corresponded to the time intervals when mice directed their nose <1 cm from the tip of the holder. Mice exploring

less than a total of 5 s within the 5 trials were excluded from the analysis and each animal was tested only once with a single odor. The odor threshold was defined as the odor concentration the most explored (relative to odorless mineral oil) among the 4 odor concentrations.

**Spontaneous odor aversion task.** Mice were kept in their home cage, water-deprived, and habituated to receiving two bottles of water for 1 h/day for 3 days. Then, a simultaneous choice was performed between one bottle containing a scented solution, either iso-amyl acetate or benzaldehyde, diluted in water and one bottle with water only in order to evaluate spontaneous odor aversion. Indeed, if some concentrations of iso-amyl acetate (0.05%) and benzaldehyde (0.01%) solutions are well accepted and considered neutral for the mice when presented alone[31], animals prefer to consume water over these scented solutions when presented a choice[58,59], this allows an evaluation of odor detection. Based on this protocol, we performed two sets of experiments using different concentrations of the odorized solution. The "highest concentrations" (iso-amyl acetate−0.05% and benzaldehyde−0.01%, Sigma-Aldrich) represent concentrations known to be perceived and equally consumed by mice and previously used in odor-conditioning protocols[31,59]. The "lowest concentrations" consist of the same solutions diluted 100 times (i.e., iso-amyl acetate−0.0005% and benzaldehyde−0.0001%). In order to promote the choice and decrease the random consumption of a solution, the odor used and the position (left or right) of the odorized solution were randomly assigned, and the top of the bottle lids were spaced about 3 cm from each other.

**Explorative odor habituation and odor discrimination task.** The protocol for explorative odor habituation and discrimination assays was conducted as previously[33] with similar material, habituation to the cage, and water deprivation prior to the test (cf Odor detection task above). For the test, mice were first presented with oil; then they were presented with two successive trials with the same odor (odor 1 → odor 1', to test habituation), followed by a presentation with a different odor (odor 2, to test discrimination). Mice that failed to explore odor 1 for more than 5 s were excluded from the analysis. For each treatment, the odor concentrations presented during the habituation and discrimination tests were based on the concentrations that mice investigated the most during the explorative odor detection task. Therefore, Vehicle treated mice were given with iso-amyl acetate and benzaldehyde concentration at 0.1% and AM251-treated mice at 0.001%.

**Calcium imaging using fiber photometry.** A fiber photometry system was used as previously described[53]. GCaMP6f was excited continuously using a 473 nm DPSS laser (output fiber intensity, 0.1–0.2 mW; Crystal Lasers) reflected on a dichroic mirror (452–490 nm/505–800 nm) and collimated into a 425 μm multimode optic fiber (NA 0.48) with a convergent lens (f: 30 mm). The emitted fluorescence was collected in the same fiber and transmitted by the dichroic mirror, filtered (525 ± 19 nm), and focused on a NewFocus 2151 femtowatt photoreceptor (Newport; DC mode). Reflected blue light along the light path was also measured with a second amplifying photodetector (PDA36A; Thorlabs) to monitor light excitation and fiber coupling. Signals from both photodetectors were digitized by a digital-to-analog converter (Power 1401; CED) at 5000 Hz and recorded using Spike2 software. Mice were progressively habituated to the bilateral connection of two flexible optical patchcord cables (Doric Lenses Inc., Quebec) within their individual home cage and within the recording cage (plastic ventilated cage; 0.5 L). Bilateral acute injections were performed via a pump through an implanted guide cannula (injection volume, 0.5 μL; speed, 0.2 μL/min via a 33-gauge cannula connected to a 10 μL Hamilton syringe) in awake animals in their home cage. After injection, animals were left to recover in their home cage for 15–25 min before being connected to the patchcord and moved to the recording chamber. After being plugged in, animals were left to recover from handling

stress in the recording cage for ~5 min before starting recordings. Spontaneous activity was monitored for 10 min by measuring raw fluorescence signals that were normalized (ΔF/F) to the mean fluorescence (50 s window), smoothed (0.02 s window), and filtered (0.02 Hz high-pass filter). Spontaneous events above 3 standard deviations (SD) were isolated and the mean frequency was calculated during the 10 min post-injection. For odor presentation, mice were placed in a small, ventilated cage (~0.5 L) coupled to a custom-built air-dilution olfactometer. Pure monomolecular odorants (iso-amyl acetate, benzaldehyde) were diluted at 10% in mineral oil in an odorless vial and saturated odor vapor was then mixed with air (dilution 1/5) before delivery into the ventilated cage (exhaust ventilation; 0.2 L/s) at a flow rate of 3 L/min. Odors were presented sequentially (4 s presentation; exhaust ventilation switched off during odor presentation) with 3 consecutive presentations of the same odor every 60 s. Global odor presentation dynamics in the cage were monitored constantly using a mini-PID (Aurora Scientific). To evaluate odor-evoked responses, we extracted the mean fluorescence during odor presentation (4 s period starting 1 s after odor onset) and normalized (ΔF/F) to the fluorescence level during the baseline period (4 s) before odor. The three consecutive odor presentations were averaged per individual. Four out of 10 mice were injected with both Vehicle and AM251, whereas 4 and 6 animals received single injections of Vehicle and AM251, respectively. The statistical unit used was the brain hemisphere.

**Olfactometer-based odor preparation and presentation.** Odors were mixed and delivered using the Aurora 220A Olfactometer. Iso-amyl acetate and benzaldehyde solutions were first manually diluted in mineral oil (highest concentration−1% or 0.2%, middle concentration− 0.1% or 0.02%, and lowest concentration 0.01% or 0.002% depending on the session) and further diluted by a factor of 10 due to the carrier flows of the olfactometer. Odors were presented each 30 times for 2 s, followed by a 20 s inter-stimulus interval during which the next odor was bubbled and air was passed through the olfactometer to clean the residue from the prior presentation. The odor delivery was calibrated using the Aurora miniPID 200B. Respiration cycles were monitored through a nasal cannula connected with a tygon tube to a pressure sensor (SSCSRNN004NDAA5, Honeywell) as in prior studies[55]. The pressure signal was recorded in parallel with electrophysiological data on one of the analog data acquisition channels.

### Histological verifications

Mice were anesthetized with ketamine/xylazine (100 mg/kg and 20 mg/kg, respectively) and perfused with 4% paraformaldehyde at the end of fiber photometry and in vivo electrophysiology experiments. Coronal sections (40–80 μm) were cut using a vibratome (Leica, VT1200S) and collected in phosphate-buffered saline (PBS). Electrode positions were examined using a donkey anti-mouse secondary antibody coupled to Cy5 (1:200, Jackson ImmunoResearch, reference 715-175-150) or to an Alexafluor 647 (1:200, Thermo Fisher Scientific, reference A31571), diluted in PBS (2 h incubation with gentle agitation), followed by a 4′,6-diamidino-2-phenylindole (DAPI) counterstaining (10 min incubation, 1:10000, Molecular Probes). Sections were washed and mounted in Fluoromont medium (Invitrogen) before being imaged with an epifluorescence microscope (Eclipse Ni-U, Nikon). For fiber photometry, fixed brains were post-fixed for 16 h in PFA 4% and then cut on a freezing microtome (Leica). Sections were washed and incubated in PBS containing 0.2% Triton and DAPI (1:10000, Molecular Probes). Sections were washed and mounted under a coverslip before being visualized with an epifluorescence microscope (Zeiss, AxioPlan 2) to validate the correct position of the cannula, fiber, or electrodes, as well as the correct GCaMP6f expression. For behavioral experiments, cannula positions were verified with neutral red staining according to manufacturer instructions[31,33] and visualized with a light microscope (Olympus, SZX2-ILLT). Animals in which post-hoc histological

examination showed that viral injection or implanted optic fiber, electrodes, or cannulas were mislocated were excluded from further analysis.

## In vivo electrophysiological recordings and analysis

All analyses were performed using Matlab (The MathWorks) built-in functions, the FMAToolbox (http://fmatoolbox.sourceforge.net/), other code developed in the Buzsáki Lab (https://github.com/buzsakilab/buzcode) and custom-written scripts.

Signals were acquired continuously at 20 kHz with an Intan RHD2000 interface board and 32- and 64-channel digital headstages (Intan Technologies). Data were visualized with Neuroscope[79] (Neurosuite, http://neurosuite.sourceforge.net). Local field potential (LFP) signal was obtained by downsampling raw data to 1250 Hz for analyses involving LFP signals.

*Pre-processing—spike sorting and unit classification.* Spike sorting was performed semi-automatically with KiloSort[80] (https://github.com/cortex-lab/KiloSort) followed by manual cluster curing in Phy (phy 2.0 beta; https://github.com/cortex-lab/phy) with the help of custom-designed plugins from Peter Peterson (https://github.com/petersenpeter/phy-plugins).

Putative excitatory and inhibitory neurons were separated on the basis of a Gaussian-mixture model using two waveform features: trough-to-peak and spike width[41] (defined hereafter as the "Cell Classifier" method; see Supplementary Fig. 1a). Only units showing a high classification confidence ($P \leq 0.01$) were used for the study and the remaining units were excluded from the analysis (49 ambiguous neurons). A total of 717 putative excitatory neurons and 102 putative inhibitory neurons were recorded in the aPC across 17 and 11 sessions from 9 mice and 8 mice for Vehicle and Rimonabant, respectively. This unit classification was confirmed using detection of putative monosynaptic connections[42,43]: 355 out of 359 cells were classified correctly as compared to the Cell Classifier method (336 excitatory and 23 inhibitory cells) and 2 excitatory and 2 inhibitory out of 359 cells were classified as ambiguous with the Cell Classifier method (see Supplementary Fig. 1). Only cells with a minimum firing rate above 0.5 Hz were kept for the following analyses: cross-correlation, inter-spike intervals and spike-LFP coupling. For olfactometer experiments, 207 putative excitatory neurons were recorded in aPC across 3 sessions from 2 mice for Vehicle and Rimonabant, respectively. A fourth session was included for the gamma oscillation analysis (Supplementary Fig. 9). This session could not be considered in the previous analysis on unit responses to odor presentations as it requires the detection of inhalation and onsets, and the respiration signal was lost in this session.

In freely moving conditions, all analyses started 10 min after each injection and comparisons between different treatments were assessed for the following 30 min. Brain states were scored based on accelerometer (movement) signals as well as hippocampal and piriform cortex spectrograms. We used the ratio of the power in theta band (5–11 Hz) to the delta band (1–4 Hz) of LFP, followed by manual adjustment with the aid of visual inspection of whitened power spectra and the raw traces (using TheStateEditor[81] from the buzcode). Only the wake state was considered for the analyses. For the olfactometer experiments, recordings also started 10 min after injection.

*LFP power and Spike-LFP coupling.* For consistency across recordings, LFP power analysis was performed on the channel which displayed the maximum number of waveforms of putative excitatory units in the aPC (waveforms are assigned according to their maximum amplitude), which presumably corresponds to layer II. If several channels displayed identical numbers of unit waveforms, the channel with the maximum power in gamma (30–80 Hz) was selected. Oscillations were extracted from LFP signals using a wavelet transform (*wavelet* function) and 8 frequency bins within the range considered

(4–12 Hz for theta; 12–30 Hz for beta; 30–80 Hz for gamma). For each 4 ms time bin, the phases and amplitudes of the oscillations were calculated from the frequency bin that showed the maximal power at any given time point. Gamma epochs used for current source density (Supplementary Fig. 3a) were extracted based on a threshold of 2 SD of the wavelet-computed gamma power (see below).

The phase preference of spikes with regards to specific LFP oscillations (Theta, 4–12 Hz; Beta, 12–30 Hz; Gamma, 30–80 Hz) was calculated for individual units by determining the distribution of spikes across 100 different phase-bins. The angular mean and resultant vector length were calculated for each unit using the Circular Statistics Toolbox[82]. In order to minimize possible confounds due to the different durations of analyzed epochs (Pre- vs Post-injection, with wake only), data were down-sampled to match the shortest interval involved in the comparison. The means (mean angle and mean resultant length) of 100 rounds of down-sampling were kept for comparison. Units were considered as phase-locked to the oscillation (i.e., modulated) based on the Rayleigh test for non-uniformity ($P < 0.05$).

To display average gamma band spectrograms from LFP oscillations recorded during odor presentations (Supplementary Fig. 9), the *spectrogram* command from the signal module in *scipy* was used with a Hamming window of 1024 samples and an overlap of 512 samples. To quantify differences, peak gamma powers (the max amplitude between 30 and 80 Hz) were extracted from each 2-s period (baseline or odor-presentation) and then averaged first across a session, then across sessions. To assess significance, Vehicle and Rim trial labels were shuffled 1000 times, repeating the peak extraction and averaging procedure described above. The distribution of average differences (Rim-Veh, in dB) derived from the shuffles was used to assess if the real differences were significant at a level of $P < 0.001$.

**Population event detection.** Multiunit synchrony events (population events—PopEvents) were identified based on previous findings[44] and detection method[83]. PopEvents were detected in sessions that included at least 10 putative excitatory units. For each session, the spike trains of all recorded excitatory units were combined, binned in 1 ms bins, and convolved with a Gaussian kernel (60 ms width, 10 ms standard deviation—SD). PopEvents intervals were identified using three criteria: (1) the multiunit firing rate deviates from at least 3 SD of the mean firing rate during Pre- and Post- injection intervals of wake state; (2) two events are separated by more than 50 ms; (3) the length of the event is between 50 and 500 ms.

**Cross-correlations and Inter-spike intervals.** Cross-correlations were computed for pairs of units using the cross-correlogram function (*ccg* function from FMAToolbox). Correlations were calculated during a 1 s window before and after reference spikes for each bin of 10 ms. Cross-correlograms were normalized by the asymptotic mean firing rates of both units.

For each unit, we computed the distribution of the inter-spike intervals (ISI) over a window of 1 s. Normalization was achieved by dividing the counts in each 1 ms bin by the total sum of the counts over the 1 s window analyzed.

For both of these analyses, only units with a minimum firing rate of 0.5 Hz were included.

**Firing rates, signal correlations, and noise correlations during olfactometry.** Putative excitatory unit spiking activity was aligned to the first inhalation in the presence of odor. Inhalation onsets were determined using a modified version of the BreathMetrics Toolbox[84]. Spikes were binned at 10 ms resolution (binning at 5 ms or 25 ms did not change the results), and firing rates (Fig. 4f and Supplementary Fig. 6) were computed as trial averages in the 350 ms window following the first inhale onset in odor. Baseline rates were assessed by performing the same analysis but in the 2-s window that occurred 1 s

before odor onset (starting 3 s before odor delivery and ending 1 s before odor delivery). As for the odor breaths, only the first breath in this window was considered for each trial to compute baseline firing rates. Significant odor responses were detected using surprise analysis[85,86]. For signal correlations, we computed Pearson's correlation coefficient for the trial-averaged firing rates of pairs of neurons. For noise correlations, we first subtracted the trial-averaged firing rates of each neuron from the responses of this neuron in each trial. The result of these subtractions was defined as the "residual" responses. Then, we computed the Pearson's correlation coefficient for the residuals of each pair of neurons on each trial. We then averaged these values across trials to obtain the noise correlation for each pair of neurons.

**Current source density.** Current source density (CSD) was computed by standard methods (available in the buzcode) after aligning LFP signals on the center of gamma epochs. Gamma epochs were detected by computing the gamma power over the whole signal (wavelet transform) on the channel with the maximum number of excitatory units (reference channel). Gamma epochs were defined by selecting time intervals when the smoothed power was above 2 SD of the overall signal power. The alignment of LFP signals was further refined by detecting the gamma oscillation trough (recorded on the reference channel) the closest to the center of each gamma epoch.

### Statistical analyses

Statistical details are presented in Supplemental Tables 1 and 2. Statistical analyses, confidence intervals, and percentile calculations were performed using Matlab (version 2018b) except for ANOVAs, where GraphPad Prism (version 9.1.0) was used. "Change" was calculated by subtracting the Post-injection from the Pre-injection measures. Comparisons were performed using nonparametric Mann-Whitney rank sum tests or Wilcoxon signed rank tests in the two-tailed configuration as stated in figure legends. Two-way ANOVA tests were chosen when appropriate (e.g., LFP frequencies, Figs. 2b and 3b, or odor concentrations, Fig. 5b, d, e and Supplementary Fig. 11b, i, j). For *post-hoc* tests, Bonferroni corrections were applied to account for multiple comparisons. Differences in proportions were assessed using Chi-square tests (Fig. 5c and Supplementary Fig. 11c and for the proportion of modulated units for spike oscillation phase-coupling analysis). Linear regressions were conducted using Spearman's correlations and tested using a Student's *t* distribution (Matlab *corr* function). Significance was set with alpha = 0.05 and was represented on graphs as the following: *$P < 0.05$, **$P < 0.01$, ***$P < 0.001$. Boxplots and data values represent median and 25–75th percentiles ([lower bound upper bound]) unless stated otherwise in the figure legends. For panels with boxplots, median, maxima, minima, 95% confidence interval ([lower bound upper bound]) of the median and 25–75th percentiles are indicated in the Source Data file.

For olfactometer data, trial-averaged baseline firing rates for each unit in each time bin aligned to inhalation onset (during the 2 s period beginning 3 s before odor delivery) were used as the rate parameters for Poisson distributions. Using these rate parameters, the survival function (tests for odor-evoked activation), and the cumulative distribution function (tests for odor-evoked inhibition), we computed the probability of the odor-evoked firing rates observed in these same bins and turned this probability into a surprise value at each time ($-\log(p)$). Surprise values were then summed across the 350 ms after inhalation onset for odor trials and mineral oil trials separately, after which the mineral oil cumulative surprise was subtracted from the odor cumulative surprises to obtain a single surprise value for a unit-odor combination. The cumulative surprise significance thresholds were set by shuffling the trial labels and rerunning this analysis 100 times for each odor for each unit (207 units, 6 odors, for a total of 124,200 points in the null distribution). With these shuffled surprises, the threshold was set at a 5% false discovery rate, meaning >95% of the shuffles. Cumulative surprises above this threshold were considered significant.

### Reporting summary

Further information on research design is available in the Nature Portfolio Reporting Summary linked to this article.

## Data availability

The data generated in this study and used to produce the figures are provided in the Source Data file. The raw physiology data are available under restricted access as they are still in use at the time of the publication. Data will be made available on request to the corresponding author. Source data are provided in this paper.

## Code availability

Most of the scripts used in this study are publicly available, as stated in the Methods. These include KiloSort (https://github.com/cortex-lab/KiloSort) and Phy 2.0 (https://github.com/cortex-lab/phy) for spike sorting, the FMAToolbox (http://fmatoolbox.sourceforge.net/) and other code developed in the Buzsáki Lab (https://github.com/buzsakilab/buzcode) for LFP and spike analyses. The BreathMetrics Toolbox from the Zelano Lab (https://github.com/zelanolab/breathmetrics) was adapted to extract inhalation onsets. Other scripts will be made available on request to the corresponding author.

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

## Acknowledgements

We thank all the staff of the Animal Facility of the Broca Institute (Pôle In Vivo) and the NeuroCentre Magendie for mouse care. The microscopy was done in the Bordeaux Imaging Center, a service unit of the CNRS-INSERM and Bordeaux University, a member of the national infrastructure France BioImaging supported by the French National Research Agency (ANR-10-INBS-04). The help of Sébastien Marais is acknowledged. We thank the Zelano lab past and current members of the Buzsáki Lab, for making the Breathmetrics Toolbox and other scripts available. We thank GENIE project, the Janelia Farm Research Campus and James M. Wilson for sharing GCamp constructs and vectors. We are also grateful to all the members of Roux' and Marsicano's lab for useful discussions. This work was supported by CNRS (to L.R., A.N., PM L.), INSERM (to G.M.), INRAE (to G.F.), Institut Pasteur (to G.L.), Ministère de l'Enseignement Supérieur et de la Recherche (MESR to G.T.), European Research Council (MiCaBra, ERC-2017-AdG-786467 to G.M. and SociOlfa ERC-StG- 851560 to L.R.), Bordeaux University (2017 IdEx Junior Chair to L.R.), Conseil régional Nouvelle-Aquitaine (Bordeaux Neurocampus Junior Chair to L.R.), Fondation pour la Recherche Médicale and Fondation Schlumberger pour l'Education et la Recherche (FSER202112014572 to L.R. and FDT20170436845 to G.T.), French National Research Agency ("ORUPS" ANR-16-CE37-0010 to G.M., G.F., and A.N.). The Perception and Action unit is supported by Institut Pasteur, CNRS and AG2R-La-Mondiale.

## Author contributions

G.T., G.M., and L.R. designed research; G.T. performed freely moving in vivo electrophysiology; G.T. and L.R. analyzed freely moving in vivo electrophysiological experiments; G.T., Y.W., D.H., and T.D. performed behavioral tasks; T.D. and L.R. performed LFP recordings with local pharmacology; G.C. adapted the scripts to analyze respiration signals; E.H. performed and analyzed head-restrained olfactometry; G.L. performed and analyzed calcium photometry; G.L., A.N., P-M.L., G.F., G.M. and L.R. supervised research; G.T. and L.R. wrote the paper. All authors edited and approved the paper.

## Competing interests

The authors declare no competing interests.
