## [Peer Review File · Nature Communications]

Endogenous cannabinoids in the piriform cortex tune olfactory perceptionREVIEWER COMMENTS

Reviewer #1 (Remarks to the Author):

This comprehensive study, the Roux group applied a combination of pharmacology, and multi-electrode single unit recordings and fiber photometry recordings in the anterior piriform cortex (aPC) of freely moving mice To test the impact of endocannabinoids on aPC network activity and on odor detection abilities.

They first show that endogenous release of cannabinoids tend to dampen the firing rates of individual aPC principal excitatory neurons and affect their firing patterns at short-time scales. Then they show that the endocannabinoid system promotes co-activation of excitatory cells specifically during population events (PopEvents), contributing to neuronal coordination in the aPC. Furthermore, they show that gamma activity and neuronal synchrony in PopEvents are linked in the aPC, suggesting that these two functions are regulated by a common mechanism involving endocannabinoid signaling, and that the endocannabinoid system is physiologically active within the aPC where it regulates the emergence of presumed bursts of activity in the local network.

Finally, they demonstrate convincingly that endocannabinoids modulate olfactory perception in the aPC by increasing odor detection thresholds.

The authors should be commended for this well planned and executed complex study. I agree that is indeed the first time that the physiological activity of CB1Rs have been shown to play a role in the fine temporal coordination of intact neuronal circuits in vivo. Their work enhances significantly our understanding not only of the role endocannabinoids on olfactory information processing. but also on network activity and olfactory perception.

I have a minor comment regarding the discussion.

In an attempt to suggest how gamma oscillations and short-time scale neuronal coordination are linked by a common mechanism in the aPC, the authors build on previous work by the Hasselmo lab, which showed that acetylcholine has a strong effect on glutamatergic synaptic transmission in the intrinsic fibers, connecting PC pyramidal neurons, but has no effect on the afferent excitatory inputs, coming from the olfactory bulb. They offer that while gamma oscillations would be a reflection of the external inputs from the olfactory bulb, PopEvents would instead result from the recurrent excitation within the aPC. They speculate that inhibition plays a key role in modulating the balance between these two phenomena, providing a powerful mechanism able to switch the system in favor of one or the other.

I am not sure if these two sets of findings can be linked together as the authors suggest. The ground breaking work of Hasselmo was conceived as a model of associative memory function in piriform cortex. Afferent and intrinsic excitatory synaptic inputs do not necessarily compete with each other, they are combined with different strengths at different times. When discussing the relation between afferent and intrinsic inputs, the authors may wish to refer to recent work coming from the Schiller lab. They may find that results described in these papers would help them in crystalizing their hypothesis regarding the connection between cellular and sub-cellular events and population activity.

Reviewer #2 (Remarks to the Author):

The manuscript titled "Endogenous cannabinoids in the piriform cortex tune olfactory perception" by Terral et al. examined the influence of CB1R blockade in anterior piriform internal network dynamics by studying gamma oscillation and population synchronization using tetrode recording and fiber cytometry. The functional role of CB1R blockade was tested with behavioral odor detection experiments. The main findings are CB1R blockade increases gamma amplitude, reduces population synchronization (Popevents and co-firing), and lowers odor detection threshold. The conclusions are the endogenous cannabinoid is tonically active in the anterior piriform and plays a role in dampening

olfactory sensory response. While the topic is interesting, techniques used are proper and results are well presented, I have several concerns.

1. Some results need to be reconciled and discussed with the previous results. Previous research (including some of the authors' work) showed CB1Rs are mainly expressed in the axon terminals of GABAergic neurons. Ex vivo activation of CB1 reduces inhibition to semilunar cells which could directly excite pyramidal cells (Terral et al 2019). Here it seems CB1 blockade also enhances excitatory neuron spiking (Fig. 1). What could explain the same excitatory effects by opposing CB1 activation or antagonism?
2. Previous research (Soria-Gomez et al. 2014) reported CB1 activation also decreases cortical feedback to the MOB and enhances odor detection. Would this contradict the enhancing effect of CB1 blockade on odor detection in the current study? Would centrifugal terminal CB1 in the MOB influence gamma oscillation in the PC?
3. The recording was conducted without odor stimulation, while the results were used to explain the functional roles of odor detection, the effect of CB1 blockade should be tested in the presence of odor to infer sensory evoked state. The cited Hasselmo paper for comparison also reported evoked synaptic responses.
4. It would be nice to acquire some information or discuss previously published results on the drug concentration and effects. A single concentration of rimonabant was used i.p. in this study. Do authors have information on blood/brain drug concentration, kinetics? How would the brain concentration via i.p. compare to the brain infusion concentration? Is there evidence to show the scale of drug diffusion following aPC infusion? Is that largely confined to aPC?
5. The authors had interesting hypothesis that CB1R blockade favors odor detection at the expense of memory, or odor discrimination, identification or categorization. While this is feasible speculation, some of these can be easily tested behaviorally. E.g. odor discrimination test. The behavioral results feel thin focusing on one aspect.

Reviewer #3 (Remarks to the Author):

- What are the noteworthy results?

This work examines the effects of a CB1 receptor antagonist on aspects of temporal organization in olfactory cortex neural activity and on olfactory perception. Authors report that systemic CB1R antagonist administration causes:

- A very subtle change in the distribution of firing rates and interspike intervals for piriform cortex excitatory neurons, with no accompanying change in inhibitory rates.
- An increase in LFP gamma oscillations and gamma entrainment of excitatory neuron spiking in piriform cortex.
- A decrease in population spiking events and pairwise coordinated increases in firing on short timescales.
- A correlated increase in LFP gamma and decrease in short-timescale spiking coordination centered around population events.

The authors carry out additional experiments to localized the drug effects to piriform cortex and develop a novel method for assessing population spiking events with fiber photometry.

Finally, the authors present behavioral evidence from two paradigms that suggest animals perceive low concentration odorants differently after systemic injection of CB1R antagonists.

The results are interpreted descriptively. The overall conclusion is that "endocannabinoids control olfactory perception by tuning population co-firing events in the aPC". Further, the authors suggest that their recordings and CSD analysis provide evidence for a specific mechanism of gamma generation within piriform cortex. Additional speculative discussion suggests that endocannabinoids increase the balance of intrinsic (recurrent) over extrinsic (afferent) control of piriform cortex population activity, contrasting with the role of cholinergic modulation.

- Will the work be of significance to the field and related fields? How does it compare to the established literature? If the work is not original, please provide relevant references.

This work provides useful information about the physiological effects of CB1 antagonism, which may contribute to our greater understanding of neuromodulation more generally in piriform cortex. The significance and importance of this factor for olfactory sensory processing is unclear. The authors do not claim to have uncovered a novel organizing principle of neuromodulation or olfactory behavior that would dramatically alter the field's focus or overall working model, but instead have contributed knowledge about a specific neuromodulator's effect on temporal organization of spiking in a specific circuit. I am not aware of prior work that would reduce the novelty of the present findings.

- Does the work support the conclusions and claims, or is additional evidence needed?

The authors do not make excessively bold claims about the effects of CB1 antagonists on spiking activity. However, convention constrains their interpretation to describing these effects as significant even though they are barely distinguishable by eye. I am not aware of any policy on effect size reporting for this journal, but this may be an opportunity to apply a tool such as Cohen's D to convey not only whether there is an effect but also some estimate of its magnitude.

The methods for identifying E and I cells from waveforms in hippocampus and neocortex are well-established with validation from alternate approaches. From the cited Stark et al: "Units were classified into putative PYR or INT based on short-latency features in the cross-correlation histograms (PYR/INT-tagging; Barthó et al., 2004), responses to brief light pulses (PV-tagging), or waveform features (Gaussian-mixture model-based classifier, trained by the cross-correlation and PV-tagging data)." It is likely that the E and I cells are identified more or less correctly from this method, but this approach has not been validated in the piriform cortex and it is a weakness of this analysis that no validation was attempted. Optogenetic tagging would be beyond the scope of this work, but the authors have the data in hand already to examine cross-correlation histograms for their identified units. This validation work ought to be reported. Also, given the long history of waveform-based cell-type identification, it is unclear why the authors should have chosen the Stark et al. reference over the primary Barthó et al. reference.

- Are there any flaws in the data analysis, interpretation and conclusions? - Do these prohibit publication or require revision?

- Is the methodology sound? Does the work meet the expected standards in your field?

- Is there enough detail provided in the methods for the work to be reproduced?

The analyses of electrophysiological recordings are standard in the field and are applied appropriately.

There is no specific reason to disbelieve the fiber photometry data, but a more rigorous study would have validated this approach against simultaneous LFP recordings to observe the correspondence of SWR-like events and calcium transients. Presumably the shift to fiber photometry stems from the difficulty of maintaining unit recordings during drug infusion, but the effect of locally infused CB1R antagonist on gamma oscillations could have been addressed with standard LFP electrophysiology.

The behavior is poorly described and, therefore, difficult to interpret. This is compounded by the fact

that exploratory behavior shows an unexpected pattern of results.

Specifically regarding behavior:

1. I cannot parse the sentence describing the odor threshold: "The odor threshold was defined as the odor concentration the most explored or the lower odor concentration for which the mice spent less than 10% to investigate it as compared to the most explored concentration."
2. If the drug effect is to lower the detection threshold, one would expect the entire investigation-dose curve to shift to the left. Instead a single concentration inspires much investigation and the others are investigated less than in the control group. This may relate to the order of presentation. Perhaps the authors really did increase the concentration from trial to trial starting from the lowest. Then the lower investigation at higher concentrations would be explained by a familiarity or habituation effect. Alternatively, the drug effect could be driven by a small number of animals that for unexplained reasons chose to explore more in the low concentration session. Readers could assess this possibility if given access to individual behavioral datapoints as are available for the water-preference task. This methodology for the odor exploration needs to be clarified, perhaps schematically. In any case, a randomized order of concentration presentation or using a single concentration per session would have avoided the confusion or confounding between behavioral habituation and detection effects.

- Notes:

1. Reference 12 is incorrectly formatted. First author initials missing.
2. Certain topics (the lack of thalamic inputs, the distinction between anterior and posterior piriform) arise in the introduction that don't bear specific relevance to the work or come up again.
3. Certain statements in the introduction are too broad or lack substance.
 - a. ". the function of other neuromodulators only start to be studied". There is a grammatical error here. More importantly, there is extensive literature on neuromodulation in piriform cortex by serotonin, norepinephrine, and adenosine as well as hormonal modulation.
 - b. "One of the most widespread and important neuromodulatory systems in the brain is the endocannabinoid system." There is no metric or method that could support this statement.
4. The section on CB1R effects in the hippocampus (83-86) could be made more specific. How is neural coordination affected? This bears on our predictions for the effects in piriform cortex.
5. The use of Antago throughout the manuscript is an unusual choice. More commonly, the drug name or an abbreviation of the drug name is used.
6. The use of "Few" and "However" in lines 78-81 seem potentially inappropriate. What is "However" contrasting with?
7. Though not an absolute requirement, the single most impactful additional experiment relevant to the authors' model would be to record neural responses to odor in piriform cortex with or without CB1R antagonist. This would create the link between network state, receptiveness to afferent input, and behaviorally-observable odor detection. An additional improvement would be to carry this recording out in animals engaged in the odor exploration task.

Reviewer #1 (Remarks to the Author):

This comprehensive study, the Roux group applied a combination of pharmacology, and multi-electrode single unit recordings and fiber photometry recordings in the anterior piriform cortex (aPC) of freely moving mice To test the impact of endocannabinoids on aPC network activity and on odor detection abilities.

They first show that endogenous release of cannabinoids tend to dampen the firing rates of individual aPC principal excitatory neurons and affect their firing patterns at short-time scales. Then they show that the endocannabinoid system promotes co-activation of excitatory cells specifically during population events (PopEvents), contributing to neuronal coordination in the aPC. Furthermore, they show that gamma activity and neuronal synchrony in PopEvents are linked in the aPC, suggesting that these two functions are regulated by a common mechanism involving endocannabinoid signaling, and that the endocannabinoid system is physiologically active within the aPC where it regulates the emergence of presumed bursts of activity in the local network.

Finally, they demonstrate convincingly that endocannabinoids modulate olfactory perception in the aPC by increasing odor detection thresholds.

The authors should be commended for this well planned and executed complex study. I agree that is indeed the first time that the physiological activity of CB1Rs have been shown to play a role in the fine temporal coordination of intact neuronal circuits in vivo. Their work enhances significantly our understanding not only of the role endocannabinoids on olfactory information processing but also on network activity and olfactory perception.

We thank the reviewer for this positive appreciation of our work.

I have a minor comment regarding the discussion.

In an attempt to suggest how gamma oscillations and short-time scale neuronal coordination are linked by a common mechanism in the aPC, the authors build on previous work by the Hasselmo lab, which showed that acetylcholine has a strong effect on glutamatergic synaptic transmission in the intrinsic fibers, connecting PC pyramidal neurons, but has no effect on the afferent excitatory inputs, coming from the olfactory bulb. They offer that while gamma oscillations would be a reflection of the external inputs from the olfactory bulb, PopEvents would instead result from the recurrent excitation within the aPC. They speculate that inhibition plays a key role in modulating the balance between these two phenomena, providing a powerful mechanism able to switch the system in favor of one or the other.

I am not sure if these two sets of findings can be linked together as the authors suggest. The ground breaking work of Hasselmo was conceived as a model of associative memory function in piriform cortex. **Afferent and intrinsic excitatory synaptic inputs do not necessarily compete with each other, they are combined with different strengths at different times.** When discussing the relation between afferent and intrinsic inputs, the authors may wish to refer to recent work coming from the Schiller lab. They may find that results described in these

papers would help them in crystalizing their hypothesis regarding the connection between cellular and sub-cellular events and population activity.

Our hypothesis was based on the inverse correlation between the changes in gamma power (presumably reflecting external inputs from the MOB) and the rate of popEvents (presumably internally generated) (Fig.3 of the initial manuscript now in Supp Fig. 3d). Because of this inverse relationship (when one becomes more prominent, the other one undergoes the reverse trend), we speculated that afferent and intrinsic excitatory inputs might be competing in the piriform cortex.

Yet, we agree with the Reviewer that we have to change this statement because (1) gamma also relies on local network interactions within the piriform cortex as shown in our data and by others (Gonzalez et al., 2023, ref.68 in manuscript) and indeed (2) the afferent and intrinsic excitatory inputs do not necessarily compete with each other. In fact, during odor presentation, odor-selective neurons are initially driven by MOB afferences and the recurrent piriform network is subsequently recruited (Bekkers and Suzuki, 2013; Davison and Ehlers, 2011; Franks et al., 2011; Giessel and Datta, 2014). Work from the Schiller lab demonstrated the existence of non-linear synaptic interactions in piriform dendrites between inputs from the MOB and inputs from the recurrent network, showing that these two types of inputs can, indeed, act in a synergistic manner (Kumar et al., 2019). We have now revised the discussion based on our new results and this information from the literature, and changed our initial statement. We also explicitly refer to the work of the Schiller lab which illustrates this idea (Discussion section, lines 390-393).

Reviewer #2 (Remarks to the Author):

The manuscript titled "Endogenous cannabinoids in the piriform cortex tune olfactory perception" by Terral et al. examined the influence of CB1R blockade in anterior piriform internal network dynamics by studying gamma oscillation and population synchronization using tetrode recording and fiber cytometry. The functional role of CB1R blockade was tested with behavioral odor detection experiments. The main findings are CB1R blockade increases gamma amplitude, reduces population synchronization (Popevents and co-firing), and lowers odor detection threshold. The conclusions are the endogenous cannabinoid is tonically active in the anterior piriform and plays a role in dampening olfactory sensory response. While the topic is interesting, techniques used are proper and results are well presented, I have several concerns.

We thank the Reviewer for considering that the topic addressed by our work is interesting and for appreciating the relevance of our experimental approaches as well as the quality of our presentation. We are also grateful for the constructive criticisms that we have addressed in the new version of the manuscript.

1. Some results need to be reconciled and discussed with the previous results. Previous research (including some of the authors' work) showed CB1Rs are mainly expressed in the axon terminals of GABAergic neurons. Ex vivo activation of CB1 reduces inhibition to semilunar cells which could directly excite pyramidal cells (Terral et al 2019). Here it seems CB1 blockade also enhances excitatory neuron spiking (Fig. 1). **What could explain the same excitatory effects by opposing CB1 activation or antagonism?**

We thank the Reviewer for this question. Indeed, the combination of previous *ex vivo* observations with our novel *in vivo* results may seem at odds given that (1) CB1R blockade slightly increases firing rates in excitatory units (new Supplementary Fig.2a-b) and (2) activation of CB1R could also lead – in principle – to an increase of activity in piriform excitatory neurons through dis-inhibition.

In order to address this issue, we performed new *ex vivo* experiments similar to Terral et al 2019 where we found that the CB1R agonist WIN decreases the frequency of mIPSCs in semilunar neurons of the aPC (Figure 5, Terral et al., 2019). We found that incubating aPC slices with the CB1R antagonist AM251 (4uM for 10min) does not affect neither the frequency nor the amplitude of mIPSCs recorded in semilunar cells (Figure A, below). These data indicate that the slight increase in firing rates observed for putative excitatory neurons in presence of the antagonist *in vivo* (new Supplementary Fig.2) cannot be linked to a change in the strength of miniature inhibitory transmission, at least as recorded *ex vivo* in semilunar cells.

Figure A: Effect of CB1R antagonist on mIPSCs of semilunar cells of the aPC. Quantifications of mIPSCs frequency (left) and amplitude (right) under Vehicle and CB1R antagonist, AM251 (4uM) recorded in semilunar cells ($n=13$ cells from 5 mice). Paired t-test, $p=0.54$ and $p=0.15$ for frequency and amplitude, respectively. This figure is not included in the current version of the manuscript.

Combined with our *in vivo* observation that spiking in putative excitatory neurons increases in the presence of the antagonist, these new results suggest that **changes in synaptic strengths using recordings of inhibitory miniature currents *ex vivo* does not necessarily reflect an observable change in neuron firing rates *in vivo*.**

Indeed, **firing rates can be governed by other factors besides inhibitory inputs *in vivo*,** such as the overall neuromodulatory tone, the levels of intrinsic excitability and the strength of excitatory inputs. The membrane properties of neurons are different *in vivo* due to the continuous synaptic activity, which results in a decrease in the overall membrane resistance. In addition, CB1Rs are also expressed in aPC excitatory cells (though at a lower level than in inhibitory cells (see Terral et al., 2019), where they play a role in the regulation of firing rates by modulating excitatory transmission. The combination and the hierarchy of these factors is not known, which makes it difficult to reconcile *ex vivo* and *in vivo* observations in general.

We additionally would like to point out that we only report a modest increase in the firing rates *in vivo* (see answer below on the reduced effect size) and this is not a key finding of our work although it seems relevant to share this information with the community.

Because of the reasons evoked here, we do not think our new results are inconsistent with previous *ex vivo* studies as they are hardly comparable. Our work instead highlights the fact that our understanding of the role of CB1R in brain physiology *in vivo* is only starting to emerge.

2. Previous research (Soria-Gomez et al. 2014) reported CB1 activation also decreases cortical feedback to the MOB and enhances odor detection. Would this contradict the enhancing effect of CB1 blockade on odor detection in the current study?

This is another interesting point of the Reviewer. However, we do not think our new data with CB1R blockade in the piriform contradict the previous publication obtained by activating CB1R in the MOB because:

i) In the present experiments, odor detection is tested with *local* injections of the antagonist *within the aPC*, thereby very likely not impacting CB1R located in the MOB. It is possible (as in many different examples in the field) that CB1 receptors in specific brain regions and/or cell types exert functions that are different and often opposite to the ones exerted in other brain regions or cell types (see for instance Soria-Gomez et al., 2021).

ii) Consistently with this idea, CB1Rs are differently expressed in the piriform cortex and in the MOB and they can have different impact on neuronal networks. In the piriform cortex, the density of CB1Rs is higher at inhibitory synapses than excitatory ones (Terral et al., 2019) which is the opposite in the MOB (Soria-Gomez et al., 2014 Nat Neurosci, Fig. 1g-i).

iii) In Soria-Gomez et al., 2014, mice were food-deprived which is not the case in this study: it is likely that food deprivation changes the role of CB1R in the olfactory system. Indeed, fasting has been shown to change cannabinoid signaling (Crosby KM et al., Neuron, 2011).

We now discuss these points in the discussion section (lines 442 to 449).

Would centrifugal terminal CB1 in the MOB influences gamma oscillation in the PC?

This intriguing suggestion of the Reviewer would indicate the presence of a form of secondary feedback control mediated by CB1 receptors in the MOB propagating to the PC. Besides the fact that it would require further specific experiments to be tested, this interesting possibility seems rather unlikely. Indeed, previous work showed that gamma oscillations are not affected by local infusions of CB1 agonists in the MOB (Soria-Gomez et al, 2014). Moreover, our new results with local injections show that CB1R blockade within the piriform *alone* contributes to the increase in gamma oscillations. These data do not support the possibility that the increased gamma oscillations originate from the MOB.

3. The recording was conducted without odor stimulation, while the results were used to explain the functional roles of odor detection, **the effect of CB1 blockade should be tested in the presence of odor to infer sensory evoked state**. The cited Hasselmo paper for comparison also reported evoked synaptic responses.

We thank the Reviewer for this important comment. Indeed, the work of Hasselmo and colleagues addressed the role of acetylcholine in the control of evoked synaptic responses in piriform acute slices, when stimulating either the afferences from the olfactory bulb, or from the piriform recurrent inputs. We agree that addressing the role of CB1R in sensory-evoked state is highly relevant to our work and therefore **we performed a series of new experiments to study the effect of CB1R blockade during controlled presentation of odors**. Specifically, we recorded odor-evoked activity in piriform cortex in two different experimental contexts (see new Figure 4).

First, using fiber photometry and local injections of CB1R antagonist (AM251) in mice expressing GCaMP6f in excitatory neurons, we measured population calcium responses to odorants presented in the recording chamber (new Fig.4a-c). We used the two odorants that were used for behavioral assays, Benzaldehyde and Isoamyl-acetate IAA, and performed control mineral oil trials. Neuronal calcium responses were recorded and normalized to pre-odor epochs (Fig. 4b; Methods). When compared to Vehicle, **we found that AM251 infusions strongly decreased the calcium responses upon odorant presentations** but not in control conditions (Fig. 4c). Fiber photometry imaging of bulk fluorescence is thought to record coordinated/synchronous population activity (Gunaydin et al., 2014). Indeed, combined *in vivo* extracellular electrophysiological recordings and population GCaMP6f imaging showed that population GCaMP6f signals are induced by synchronous population bursts and not by single cell firing (Li et al., 2019 Front. Cell. Neurosci.). Thus, we hypothesized that the AM251-induced reduction of calcium responses is mainly due to a decorrelation of the GCaMP6-expressing neurons resulting in the decrease in population bursts.

To address this hypothesis, we conducted another set of experiments to gain cellular resolution and analyze single-cell firing. Thus, we chronically implanted silicon probes into the piriform cortex, in addition to a nasal cannula to measure respiration, and a head-post to allow head-fixation in front of an olfactometer. Head-fixed mice were presented the 2 different odors (Benzaldehyde and IAA) at 3 different concentrations while recording breathing and piriform cortex spiking activity after systemic injection of either the CB1R antagonist Rim or Vehicle. We found that **during CB1R blockade, odor-evoked spiking in the first breath after odor onset was maintained**, even though baseline firing rates taken from breaths in the inter-stimulus periods were decreased (see Supp. Fig. 6). This amounts to an increase in first-breath odor-evoked spiking relative to baseline during CB1R blockade (new Fig.4f). At the population level, **we found that piriform odor responses were decorrelated during CB1R blockade**. Signal correlations, which are correlations between the trial-averaged responses of pairs of neurons, and noise correlations, which are the residual trial-by-trial correlations after subtracting the mean responses, were both significantly decreased during CB1R blockade, and this effect was the strongest for the odor-concentration that evoked the most correlated activity in the Vehicle condition (highest concentration IAA, see new Fig. 4g and Supp. Fig. 7 and 8). These results, taken together with the fiber photometry measurements after local CB1R blockade, echo our observations in freely moving conditions: CB1R blockade appears to decorrelate cortical activity in the PC.

We are very grateful to the Reviewer for his/her suggestion, because these new data allow us refining the conclusions of our study **that CB1Rs are key players in the modulation of piriform neuronal networks also in the presence of external sensory inputs**. This new set

of results on odor-evoked responses is now included in the manuscript as a complement to our initial observations on the basal internal state.

Although the properties of odor-evoked responses are interesting, we think that the ability to detect an unexpected odorant at low concentration (e.g. in the detection task presented in Figure 5) largely depends on the internal network dynamics in baseline, that can make the circuits more “receptive” to external inputs. This point is discussed in the new version of the manuscript (lines 397 to 421).

4. It would be nice to acquire some information or discuss previously published results on the drug concentration and effects. A single concentration of rimonabant was used i.p. in this study. Do authors have information on blood/brain drug concentration, kinetics? How would the brain concentration via i.p. compare to the brain infusion concentration?

For our pharmacology experiments, we used the concentrations validated in the field (including - but not only - our previous studies) both for IP (1 mg/kg of rimonabant; Terral et al., 2019; Busquets-Garcia et al., PNAS 2016; Busquets-Garcia et al., Neuron 2018; Grim TW et al., 2017; Rodriguez Bambico et al., 2010) and local injections (AM251 4ug; Soria-Gomez et al., Nature Neurosci. 2014; Daniela da Fonseca Pacheco et al., BJP 2009; Soria-Gomez et al., Neuron 2021) of the CB1R antagonist. As proposed by the Reviewer, we include this information in the Methods section (lines 557 to 558). To our knowledge, there is no method to precisely compare the drug concentrations between IP and local infusions. The best approach remains to identify a *functional* effect of the drug and test whether it is the same with IP and local drug administrations. Here, we found that the impact of CB1R blockade on piriform population bursts is similar with the two approaches (see Fig.1e,f and Fig.3f for comparison).

Is there evidence to show the scale of drug diffusion following aPC infusion? Is that largely confined to aPC?

This important question was addressed in one of our previous publications on the impact of CB1R in the piriform cortex: Terral et al., 2019. In this former study, local infusions of the antagonist AM251 were conducted in the exact same way as in our new manuscript (same coordinates, same injection protocol, same material). The diffusion of the drug was estimated using local infusions of pontamine sky blue and showed that the compound mostly diffused in the aPC. Below is the corresponding figure (Supplementary Figure 2 in Terral et al., 2019):

Figure B: (A) Injection cannula tips in the aPC of 35 randomly selected vehicle or AM251 treated mice from all the pharmacological experiments (red circles). Adapted from Paxinos and Watson [65]. (B) Representative image showing the aPC injected site (blue) for local pharmacological treatments. Numbers in (A) and (B) indicate the relative position of coronal slices from bregma. (C) Radius spreading along the anteroposterior axis of the pontamine sky blue injected through 30 cannulas randomly selected from the different pharmacological experiments.

Despite these previous controls, it is still possible that pontamine sky blue has different diffusion properties than AM251. Indeed, this CB1R antagonist is a hydrophobic molecule, which might diffuse differently from a water-soluble dye like pontamine. Thus, to fully address the correct concern of the Reviewer, we conducted additional experiments with local infusions of the hydrophobic fluorescent dye 1,1'-dioctadecyl-3,3,3',3'-tetramethylindocarbocyanine perchlorate (DiI). The results were very similar as the previous ones obtained with pontamine, revealing that a hydrophobic molecule diffuses even less (Figure C below).

This information about test injections is now included in Supplementary Figure 4.

Figure C: Histological verification of injection sites in aPC. Four brain sections from four different mice implanted with injection cannulas above the aPC. Dil was injected through the injector 1.5 mm below the cannula's tip and the brain tissue was fixed immediately after. Because of its hydrophobic properties, diffusion of Dil beyond the injection site is limited showing the location of the tip of the injector. Scale bar: 1mm.

5. The authors had interesting hypothesis that CB1R blockade favors odor detection at the expense of memory, or odor discrimination, identification or categorization. While this is feasible speculation, some of these can be easily tested behaviorally. E.g. odor discrimination test. The behavioral results feel thin focusing on one aspect.

In the initial version of the manuscript, we reported behavioral results obtained in odor detection tasks only and used the Discussion section to share our thoughts about the role of CB1R in other types of olfactory tasks. We thank the Reviewer for his/her positive appreciation of the hypotheses proposed in our general discussion.

In order to test our hypotheses, in line with the Reviewer suggestion, we added information about other behavioral data besides the two odor detection tasks:

(1) In our previous study (Terral et al. 2019), we found that memory retrieval was impaired when CB1Rs were blocked specifically in the piriform cortex (same coordinates, same drug concentration as in the current manuscript). These results are in line with the first proposed hypothesis that CB1R blockade favors odor detection at the expense of memory and are now more explicitly described and discussed in the context of the present study (lines 458 to 463).

(2) To expand the range of the effects of CB1R blockade on olfactory-dependent processes, **we performed additional experiments with local pharmacology, testing for olfactory habituation and discrimination performances** in mice implanted with bilateral cannulas in the aPC. These data are now included in Supplementary Figure 10h-j, described in lines 303 to 323 and discussed in lines 458 to 463. Briefly, we found that odor habituation is not impacted when CB1Rs are specifically blocked in the piriform cortex (mice explore less the odor source when the same stimulus is presented twice consecutively – O1 vs O1'). Yet, odor discrimination is impaired in antagonist-infused mice as compared to vehicle-infused control mice (antagonist-treated mice explore less the new presented odor – O2 when compared to O1' – than vehicle-

treated mice). These results are in line with the proposed hypothesis that CB1R blockade favors odor detection at the expense of odor discrimination.

Although we plan to do additional experiments to understand the underlying mechanism as part of a separate study, we think that adding these new data strengthen the impact of our current manuscript by showing that CB1R is central to fine tune the balance between different cognitive processes in the piriform cortex.

Reviewer #3 (Remarks to the Author):

- What are the noteworthy results?

This work examines the effects of a CB1 receptor antagonist on aspects of temporal organization in olfactory cortex neural activity and on olfactory perception. Authors report that systemic CB1R antagonist administration causes:

- A very subtle change in the distribution of firing rates and interspike intervals for piriform cortex excitatory neurons, with no accompanying change in inhibitory rates.
- An increase in LFP gamma oscillations and gamma entrainment of excitatory neuron spiking in piriform cortex.
- A decrease in population spiking events and pairwise coordinated increases in firing on short timescales.
- A correlated increase in LFP gamma and decrease in short-timescale spiking coordination centered around population events.

The authors carry out additional experiments to localized the drug effects to piriform cortex and develop a novel method for assessing population spiking events with fiber photometry.

Finally, the authors present behavioral evidence from two paradigms that suggest animals perceive low concentration odorants differently after systemic injection of CB1R antagonists.

We thank the Reviewer for this clear summary of our work. In addition, we would like to clarify the point addressed in the last sentence (about our behavioral results): animals perceive low concentration odorants after *local* injections of CB1R antagonists within the anterior piriform cortex. This information is important as it indicates that the effect on odor perception is specific for CB1R located *within* the aPC.

The results are interpreted descriptively. The overall conclusion is that “endocannabinoids control olfactory perception by tuning population co-firing events in the aPC”. Further, the authors suggest that their recordings and CSD analysis provide evidence for a specific mechanism of gamma generation within piriform cortex. Additional speculative discussion suggests that endocannabinoids increase the balance of intrinsic (recurrent) over extrinsic (afferent) control of piriform cortex population activity, contrasting with the role of cholinergic modulation.

- Will the work be of significance to the field and related fields? How does it compare to the established literature? If the work is not original, please provide relevant references.

This work provides useful information about the physiological effects of CB1 antagonism, which may contribute to our greater understanding of neuromodulation more generally in piriform cortex. The significance and importance of this factor for olfactory sensory processing is unclear. The authors do not claim to have uncovered a novel organizing principle of neuromodulation or olfactory behavior that would dramatically alter the field's focus or overall working model, but instead have contributed knowledge about a specific neuromodulator's effect on temporal organization of spiking in a specific circuit. I am not aware of prior work that would reduce the novelty of the present findings.

We are grateful for this appreciation of the novelty of our findings and for acknowledging our contribution to our understanding of the role of CB1R in the fine coordination of neuronal circuits in the piriform cortex.

- Does the work support the conclusions and claims, or is additional evidence needed?

The authors do not make excessively bold claims about the effects of CB1 antagonists on spiking activity. However, convention constrains their interpretation to describing these effects as significant even though they are barely distinguishable by eye. I am not aware of any policy on effect size reporting for this journal, but this may be an opportunity to apply a tool such as Cohen's D to convey not only whether there is an effect but also some estimate of its magnitude.

We agree with the Reviewer that the impact of CB1R antagonist on the firing rates observed in the piriform is a significant but small effect. Accordingly, the accompanying text was reflecting this observation that we fully acknowledge: "Following CB1R Antago injections, we found a *modest*, but significant increase in E cell firing rates as compared to pre-injection and Vehicle group". To further support this claim, according to the Reviewer suggestion, we have now applied a Cohen's D analysis on the firing rate dataset. The latter reveals the small effect size with a Cohen D of 0.105. Accordingly, in the Results section, we have now changed our sentence to reflect the small amplitude of the effect: "we found that CB1R blockade increased the firing rates of E_Cells, yet with a small effect size (Cohen's D: 0.105)." (lines 133 to 135). Of note, only mild effects on neuronal firing rates were also reported in the hippocampus CA1 when CB1Rs were activated (Robbe et al., 2006).

The methods for identifying E and I cells from waveforms in hippocampus and neocortex are well-established with validation from alternate approaches. From the cited Stark et al 2013, Units were classified into putative PYR or INT based on short-latency features in the cross-correlation histograms (PYR/INT-tagging; Bartho et al., 2004), responses to brief light pulses (PV-tagging), or waveform features (Gaussian-mixture model-based classifier, trained by the cross-correlation and PV-tagging data). It is likely that the E and I cells are identified more or less correctly from this method, but this approach has not been validated in the piriform cortex and it is a weakness of this analysis that no validation was attempted. Optogenetic tagging would be beyond the scope of this work, but the authors have the data in hand already to examine cross-correlation histograms for their identified units. This validation work ought to be reported. Also, given the long history of waveform-based

cell-type identification, it is unclear why the authors should have chosen the Stark et al. reference over the primary Bartho et al. reference.

We used the Stark classifier over the Bartho et al. method because it presents the advantage of being constructed on ground-truth data from opto-tagged units and identified putative mono-synaptic connections. It can therefore provide a confidence level (p value) for each individual unit classification (if confidence is too low – $P > 0.01$, the unit is considered as ambiguous and thus not considered in our study). Such a feature is not available in the Bartho method.

While it is likely that waveform features are similar between the piriform and the hippocampus (as it is the case between neocortex and hippocampus), we agree with the Reviewer that the method used to classify putative E and I_Cells had to be further validated. To do so, we followed his/her advice and examined cross-correlation (CCG) histograms of simultaneously recorded units to detect putative monosynaptic connections as in Bartho et al., 2004 with the additional refinement proposed in the Stark and Abeles 2009: the peak in the CCG needed to exceed that from the slowly co-modulated baseline. This method has now been used in numerous studies such as Girardeau and Buzsáki, Nature Neurosci. 2017; Senzai and Buzsáki, Neuron 2019; English et al., Neuron 2017.

Based on this approach, we identified 350 putative excitatory and 26 putative inhibitory mono-synaptic connections in our dataset (see Methods, lines 722 to 726). With this information in hands, we compared the E and I_Cells identified via this method with the E and I_Cells identified with our classifier. **We found that 334 out of the 336 excitatory cells identified with mono-synaptic connections (cross-correlations) were correctly assigned** to the excitatory cells group with the classifier and **21 out of 23 inhibitory cells identified with mono-synaptic connections were correctly assigned** to the inhibitory cell group. The remaining 4 cells identified with mono-synaptic connections were assigned as ambiguous with our cell classifier. The cells identified through putative mono-synaptic connections are now displayed in Supplementary Figure 1.

Given the low error rate highlighted by this comparison, we considered our cell classification as acceptable for piriform cortex units. We thank the Reviewer for suggesting this improvement in our method.

- **Are there any flaws in the data analysis, interpretation and conclusions? - Do these prohibit publication or require revision?**
- **Is the methodology sound? Does the work meet the expected standards in your field?**
- **Is there enough detail provided in the methods for the work to be reproduced?**

The analyses of electrophysiological recordings are standard in the field and are applied appropriately.

There is no specific reason to disbelieve the fiber photometry data, but a more rigorous study would have validated this approach against simultaneous LFP recordings to observe the correspondence of SWR-like events and calcium transients. Presumably the shift to fiber photometry stems from the difficulty of maintaining unit recordings during drug infusion, but the effect of locally infused CB1R antagonist on gamma oscillations could have been addressed

with standard LFP electrophysiology.

We appreciate the suggestion of the Reviewer and we see her/his point. Indeed, unfortunately, the detection of the popEvents during local drug infusions is not possible with standard LFP electrophysiology as it relies on the detection of neuronal co-activation through unit spiking activity and drug infusion induces tissue movements that are incompatible with reliable unit recordings. The detection of sharp waves in the LFPs (independently of unit recordings) is not a good indicator of the presence of populations events as these two types of events are not always concomitant in our silicon probe recordings (see for instance Fig 1d where a population event is detected but no clear sharp wave is visible in the LFP). Fiber photometry represents a way to circumvent the issue of unit stability as it detects population calcium events and are compatible with drug infusions. We are confident that these calcium events are similar to the population events detected with spiking activity (and silicon probe recordings) as their occurrence rate is in the same range (~ 0.4 events/sec).

However, we agree with the Reviewer that gamma LFP recordings are – in principle - compatible with local drug infusion as they do not require unit stability. The potential effect of local infusions of CB1R antagonist on gamma oscillations was indeed lacking in the initial version of our manuscript. **To address this legitimate question, we performed additional experiments in a new set of 8 mice.** We set up a new challenging method which consists in attaching a tungsten wire to an injection cannula and to implant the so-formed device in the aPC to record LFP signal in the close vicinity (~500um) from the site of drug delivery. The results obtained in this new set of experiments are included in the Figure D below and in the Figure 3a,b and Supplementary Fig 4 of the new manuscript.

Figure D: Effect of local manipulation on aPC oscillations. (a) A tungsten wire is attached to the injection cannula and connected to an Omnetic connector. (b) Schematic of a cannula coupled with a 50um tungsten wire implanted unilaterally in the aPC for LFP recordings. During injections, the injector was inserted into the cannula and protruded 1.5mm below the cannula's

tip, ~500um above the tip of the tungsten wire. (c) Mean change in power spectrum induced by Vehicle (black) or AM251 (orange) injections. Two-way ANOVA interaction: *** $P < 0.001$. Horizontal line indicates post-hoc student *t* test with Bonferroni correction for multiple comparisons: * $P < 0.05$; $n = 14$ and $n = 14$ sessions from 8 mice for Vehicle and AM251, respectively. Means, bold lines; SEM, shaded areas.

In line with results obtained with systemic blockade of CB1R, local infusions of a CB1R antagonist (AM251, 4ug) significantly increased LFP power in the low gamma frequency range when compared to Vehicle injections (**Figure Dc above**). As a control, we injected tetrodotoxin (TTX, 5ug) through the same cannulas and observed a robust decrease of LFP power in all frequency bands (**Supplementary Fig. 4b**), indicating that the injection/recording system was functional and highlighting the specificity of the effect of the CB1R antagonist on gamma oscillations. These new important results now described lines 194 to 206 of the manuscript, suggest that the increase in gamma oscillations observed with systemic injections of the CB1R antagonist was likely mediated – at least in part - by CB1R located within the aPC.

We thank the Reviewer for suggesting this new experiment that consolidates our initial observations and highlights the importance of *local* CB1R in modulating gamma oscillations in the aPC.

The behavior is poorly described and, therefore, difficult to interpret. This is compounded by the fact that exploratory behavior shows an unexpected pattern of results.

We apologize for the lack of clarity and will improve the manuscript according to the specific comments below.

Specifically regarding behavior:

1. I cannot parse the sentence describing the odor threshold: “The odor threshold was defined as the odor concentration the most explored or the lower odor concentration for which the mice spent less than 10% to investigate it as compared to the most explored concentration”;

We agree that the sentence we used to define the threshold for the first concentration of odor detected was difficult to understand and we deeply apologize for this lack of clarity. We therefore decided to simplify our analysis using the following definition: “the odor threshold was defined as the odor concentration the most explored (relative to odorless mineral oil) among the 4 odor concentrations” (see Methods, lines 605 to 607). We changed the figure accordingly (now Figure 5). This modification did not change the general conclusion that mice infused with CB1R antagonist show lower thresholds (mainly 0.001%) than the vehicle-infused mice that display higher thresholds accompanied by a larger variability of threshold values (mainly 0.01% and 0.1%).

2. If the drug effect is to lower the detection threshold, one would expect the entire investigation-dose curve to shift to the left. Instead a single concentration inspires much investigation and the others are investigated less than in the control group. This may relate to the order of presentation. Perhaps the authors really did increase the concentration from trial to trial starting from the lowest. Then the lower investigation at higher concentrations would be explained by a

familiarity or habituation effect. Alternatively, the drug effect could be driven by a small number of animals that for unexplained reasons chose to explore more in the low concentration session. Readers could assess this possibility if given access to individual behavioral datapoints as are available for the water-preference task. This methodology for the odor exploration needs to be clarified, perhaps schematically. In any case, a randomized order of concentration presentation or using a single concentration per session would have avoided the confusion or confounding between behavioral habituation and detection effects.

The questions raised by the Reviewer highlight the fact that we had to clarify: (1) the protocol that was used for this odor detection task and (2) the results obtained. We provide answers regarding these 2 points below and performed a new set of experiments to confirm that CB1R blockade do not affect odor habituation (see Reviewer #2 .5).

1- Protocol:

Indeed, we used an experimental design commonly used for testing odor detection thresholds (Witt et al., J Vis Exp. 2009; Soria-Gómez et al., Nat. Neuro 2014; Yang & Crawley Curr. Protoc. Neurosci. 2009) where odor concentrations are increased from trial to trial, from the lowest concentration to the highest. Randomized order of concentrations cannot be used in this case because this method do not exclude the habituation effect that we clearly observed in our data (ie once an odor is detected, mice tend to explore it less in the following trials) (see for instance Supplementary Figure 10f,i). Testing a single odor per session would also have been problematic because the degree of novelty of the odor would differ from session to session. We think our test, done with odors the mouse has never encountered before, allows accurately assessing the lowest odor concentration for which mice show an increased interest when compared to odorless oil solution. We have clarified the methodology used in the Methods section (lines 605 to 607) and complemented our behavioral testing with habituation tests, verifying that both groups of mice show a similar level of habituation (Supplementary Figure 10i and answer to Reviewer 2 above).

2- Results:

The investigation dose-curve is not shifted to the left in the AM251-infused mice because of the dispersion of the threshold values observed in the control vehicle group. To provide a clearer representation of this phenomenon, we propose the figure below:

By splitting the data according to the threshold values of individual mice, it becomes clear that the curve for AM251-infused mice is indeed shifted towards the left and has the same shape as mice in the control groups.

Finally, we now provide individual data point for the odor detection task as requested by the Reviewer in the Supplementary Figure 10f. To improve clarity, we also split the data according to the threshold values of individual mice in that figure (the panel is also shown below).

We hope these modifications will help the reader understanding the observed phenomenon and the fact that with two different behavioral tasks and two different sets of mice, detection thresholds are consistently decreased when CB1Rs are specifically blocked in the aPC.

Notes:

1. Reference 12 is incorrectly formatted. First author initials missing.

We have corrected this mistake.

2. Certain topics (the lack of thalamic inputs, the distinction between anterior and posterior piriform) arise in the introduction that don't bear specific relevance to the work or come up again.

We agree with the Reviewer and removed the distinction between the anterior and posterior piriform in the introduction accordingly.

3. Certain statements in the introduction are too broad or lack substance.

"the function of other neuromodulators only start to be studied". There is a grammatical error here. More importantly, there is extensive literature on neuromodulation in piriform cortex by serotonin, norepinephrine, and adenosine as well as hormonal modulation.

In line with the Reviewer's comment and the review from Linster and Cleland (2016), we modified the sentence in the introduction by the following: "Besides the well-characterized regulation of piriform circuits by acetylcholine, the function of other neuromodulators in this brain region have also been studied²⁶. Yet, how the endocannabinoid system (ECS) modulates aPC circuit function in vivo remains elusive²⁷" (lines 65 to 66).

b. "One of the most widespread and important neuromodulatory systems in the brain is the endocannabinoid system." There is no metric or method that could support this statement.

We agree and changed this statement.

4. The section on CB1R effects in the hippocampus (83-86) could be made more specific. How is neural coordination affected? This bears on our predictions for the effects in piriform cortex.

We modified this part and provided additional details on the experiments done in the hippocampus: "For example, hippocampal sharp-wave ripples and unit entrainment by the theta rhythm are disrupted with CB1R manipulations (Robbe et al., 2006 - 2009)." (lines 84 to 85).

5. The use of Antago throughout the manuscript is an unusual choice. More commonly, the drug name or an abbreviation of the drug name is used.

We had chosen the wording "Antago" to facilitate the reading but we agree that this is less precise. We have done the modification as suggested.

6. The use of "Few" and "However" in lines 78-81 seem potentially inappropriate. What is "However" contrasting with?

We apologize for this mistake and have completely rewritten the paragraph (lines 79-85).

7. Though not an absolute requirement, the single most impactful additional experiment relevant to the authors' model would be to record neural responses to odor in piriform cortex with or without CB1R antagonist. This would create the link between network state, receptiveness to afferent input, and behaviorally-observable odor detection. An additional improvement would be to carry this recording out in animals engaged in the odor exploration task.

We thank the Reviewer for this important comment. To study the effect of CB1R blockade during controlled odor presentations, **we recorded odor-evoked activity in piriform cortex in two different experimental contexts**, which were chosen to maximize the control of odor presentations and the number of trials. The results of these new experiments are now included in Figure 4.

First, we introduced odorants in a hermetic chamber containing freely moving mice and used fiber photometry in combination with local injections of the CB1R antagonist in the aPC. We found a reduction in the calcium signal observed during odor presentations when CB1R were blocked in the aPC, suggesting that neuronal responses were decorrelated among the neuronal population.

In a second set of experiments, we chronically implanted mice with silicon probes in the piriform cortex, in addition to a nasal cannula to measure respiration and a head-post to allow head-fixation in front of an olfactometer. Head-fixed mice were presented 2 different odors (iso-amyl acetate and benzaldehyde) at 3 different concentrations while recording breathing and piriform cortex spiking activity after systemic injection of either the CB1R antagonist or vehicle. We found that during CB1R blockade, odor-evoked spiking in the first breath after odor onset was maintained, even though baseline firing rates taken from breaths in the inter-stimulus periods were decreased (see new Figure 4 and Supplementary Figure 6). This amounts to an increase in first-breath odor-evoked spiking relative to baseline during CB1R blockade. **At the population level, we found that piriform odor representations were decorrelated during CB1R blockade.** Signal correlations, which are correlations between the trial-averaged responses of pairs of neurons, and noise correlations, which are the residual trial-by-trial correlations after subtracting the mean responses, were both significantly decreased during CB1R blockade, and this effect was stronger for the odor-concentration that evoked the most correlated activity in vehicle condition (highest concentration iso-amyl acetate, see Figure 4 and Supplementary Figure 7-8). **These results, taken together with the fiber photometry measurements after local CB1R blockade, echo our observations in freely moving conditions: CB1R blockade decorrelates cortical activity associated with sensory-evoked states.**

REVIEWERS' COMMENTS

Reviewer #1 (Remarks to the Author):

The authors revised the manuscript nicely according to the the reviewers' comments.

Reviewer #2 (Remarks to the Author):

The authors have addressed my concerns satisfactorily.

Reviewer #3 (Remarks to the Author):

The authors introduce multiple new lines of evidence to address concerns raised in the original submission.

My most pressing concern was the difficulty of interpreting the behavioral methodology and results. This has been fully addressed in revision. The results are now clear.

I also suggested that, though not strictly necessary, recording odor-evoked responses in piriform cortex could provide a link between electrophysiological changes and behavior. The authors now report an entirely new dataset indicating that baseline rates are decreased after CB1R antagonist delivery. This could potentially explain the reduced behavioral threshold for detection.

Finally, I suggested that it should be possible to measure drug-induced LFP changes directly. The authors developed a technically challenging simultaneous recording/infusion approach to address this concern.

My assessment is that this additional work sufficiently addresses the issues raised in review.

I include the following additional comments to the authors only as suggestions:

1. The suppressive effect of local vehicle infusions on LFP seems to indicate this technique needs further development for future applications. The authors suggest that this stems from vehicle hydrophobicity or tissue distortion. The effect of hydrophobicity could be tested independently. Additional adjustments to the infusion protocol or cannula placement could improve tissue distortion effects.

2. The argument that calcium events and population events reflect the same phenomenon because their rates are the same is not strongly convincing. Each of these events are discovered after a number of signal processing steps and subject to thresholding criteria. Thus, the rate of event detection is necessarily determined by decisions made during analysis rather than a fundamental physiological phenomenon. I would advise deemphasizing this line of argument.

3. In the Discussion section on Impact of CB1R signaling on olfactory stimulus responses, the relationships between different operational definitions of coordination and correlation could be more clear.

Sentence 1: “neural coordination in baseline conditions” would seem to refer to reduced short time-scale coordination observed in Figure 1g and 3f.

Sentence 2: “population calcium responses” are also supposed to reflect short time scale synchronous population events.

Sentence 3: responses explained by “decreased signal correlation”. Signal and noise correlations are calculated over a 350 msec window. Signal correlations reflect similar odor tuning across neuron pairs.

It is not clear to me that a link is established here between odor tuning over a 350 ms window and the short time scale population events reported in Figures 1-3.

REVIEWERS' COMMENTS

Reviewer #1 (Remarks to the Author):

The authors revised the manuscript nicely according to the the reviewers' comments.

We thank the Reviewer for their appreciation and suggestions which improved our manuscript.

Reviewer #2 (Remarks to the Author):

The authors have addressed my concerns satisfactorily.

We are happy to read that the new experiments and our responses addressed the relevant points previously raised by the Reviewer.

Reviewer #3 (Remarks to the Author):

The authors introduce multiple new lines of evidence to address concerns raised in the original submission.

My most pressing concern was the difficulty of interpreting the behavioral methodology and results. This has been fully addressed in revision. The results are now clear.

I also suggested that, though not strictly necessary, recording odor-evoked responses in piriform cortex could provide a link between electrophysiological changes and behavior. The authors now report an entirely new dataset indicating that baseline rates are decreased after CB1R antagonist delivery. This could potentially explain the reduced behavioral threshold for detection.

Finally, I suggested that it should be possible to measure drug-induced LFP changes directly. The authors developed a technically challenging simultaneous recording/infusion approach to address this concern.

My assessment is that this additional work sufficiently addresses the issues raised in review.

We are happy that the new set of experiments and the modifications of our text suggested by the Reviewer are appreciated, and we are grateful for the resulting improvement in our manuscript.

Indeed, the change in baseline rates are decreased after CB1R antagonist delivery in the context of repeated odor presentations in the olfactometer and this could potentially contribute to the behavioral effect on olfactory detection threshold. For this reason, the Discussion includes the following sentence: "In parallel, the reduced noise correlation and increased evoked-responses we observed during repeated stimulus presentations in head-fixed mice could increase the reliability and the efficiency of neuronal responses to odorants." (lines 405-408).

I include the following additional comments to the authors only as suggestions:

1. The suppressive effect of local vehicle infusions on LFP seems to indicate this technique needs further development for future applications. The authors suggest that this stems from vehicle hydrophobicity or tissue distortion. The effect of hydrophobicity could be tested independently. Additional adjustments to the infusion protocol or cannula placement could improve tissue distortion effects.

We thank the reviewer for the useful suggestions. Displacing the cannula with respect to the electrode could help with the tissue distortion at the cost of requiring the drug to diffuse further before we can measure its effect.

2. The argument that calcium events and population events reflect the same phenomenon because their rates are the same is not strongly convincing. Each of these events are discovered after a number of signal processing steps and subject to thresholding criteria. Thus, the rate of event detection is necessarily determined by decisions made during analysis rather than a fundamental physiological phenomenon. I would advise deemphasizing this line of argument.

We have changed the text according to this suggestion. Lines 340 we wrote the following: “Similar to preceding studies⁴²⁻⁴⁴, our extracellular unit recordings and fiber photometry experiments highlighted the presence of PopEvents and population calcium transients in the aPC.” The sentence now replaces the preceding sentence that the Reviewer criticized: “In our study, we found that calcium transients occur at a similar rate to PopEvents, suggesting that our electrophysiological and fiber photometry experiments likely captured the same physiological phenomena.”

3. In the Discussion section on Impact of CB1R signaling on olfactory stimulus responses, the relationships between different operational definitions of coordination and correlation could be more clear.

Sentence 1: “neural coordination in baseline conditions” would seem to refer to reduced short time-scale coordination observed in Figure 1g and 3f.

Sentence 2: “population calcium responses” are also supposed to reflect short time scale synchronous population events.

Sentence 3: responses explained by “decreased signal correlation”. Signal and noise correlations are calculated over a 350 msec window. Signal correlations reflect similar odor tuning across neuron pairs.

It is not clear to me that a link is established here between odor tuning over a 350 ms window and the short time scale population events reported in Figures 1-3.

We apologize for the lack of clarity. Signal correlation is indeed calculated differently from the

synchronous population events (or population calcium responses) extracted in baseline conditions (i.e. outside of stimulus presentations). Signal correlation measures the similarity of the trial-averaged responses among pairs of neurons. In our study, signal correlation and population events are independent phenomena that could both be influenced by the degree of co-activation within the neuronal population. We have modified the Discussion to clarify this point: “Although signal correlation and PopEvents are independent phenomena (being related to odor presentations or baseline conditions, respectively), they are both influenced by the degree of co-activation within neuronal populations” (lines 401-403).